# Physics-Informed Self-Guided Diffusion Model for High-Fidelity Simulations

## Abstract

Machine learning (ML) models are increasingly explored in fluid dynamics as a promising way to generate high-fidelity computational fluid dynamics data more efficiently. A common strategy is to use low-fidelity data as computational-efficient inputs, and employ ML techniques to reconstruct high-fidelity flow fields. However, existing work typically assumes that low-fidelity data is artificially downsampled from high-fidelity sources, which limits model performance. In real-world applications, low-fidelity data is generated directly by numerical solvers with a lower initial state resolution, resulting in large deviations from high-fidelity data. To address this gap, we propose *PG-Diff*, a novel diffusion model for reconstructing high-fidelity flow fields, where both low- and high-fidelity data are generated from numerical solvers. Our experiments reveal that state-of-the-art models struggle to recover fine-grained high-fidelity details when using solver-generated low-fidelity inputs, due to distribution shift. To overcome this challenge, we introduce an *Importance Weight* strategy during training as self-guidance and a training-free *Residual Correction* method during inference as physical inductive bias, guiding the diffusion model toward higher-quality reconstructions. Experiments on four 2D turbulent flow datasets demonstrate the effectiveness of our proposed method. Code and further details are available here.

## 1 Introduction

High-fidelity simulations of computational fluid dynamics (CFD) are crucial for understanding fluid interactions in engineering systems, greatly impacting design and application outcomes (Wang et al., 2024; McGreivy & Hakim, 2024). Traditional approaches such as Direct Numerical Simulation (DNS) (Orszag, 1970) offer high-resolution solutions. However, they are computationally expensive, especially for complex dynamics such as turbulence with high Reynolds numbers (Zhang et al., 2023). Therefore, learning neural-based simulators from data become attractive alternatives, balancing between efficiency and simulation fidelity (Huang et al., 2023).

One popular strategy is to reconstruct high-fidelity data from low-fidelity inputs, which usually reduces the discretization grid size in the spatial domain to improve computational efficiency (Shu et al., 2023; Pradhan & Duraisamy, 2021). Various machine learning models, including those based on Convolutional Neural Networks (CNNs) (Fukami et al., 2019), Generative Adversarial Networks (GANs) (Li & McComb, 2022), and Diffusion Models (Shu et al., 2023), have been developed to reconstruct high-fidelity CFD data from low-fidelity inputs. Majority of them are categorized as direct mapping models, which require both low- and high-fidelity data for training and can only capture the relationship between particular low-fidelity and high-fidelity pairs. In contrast, diffusion models only require high-fidelity data during training and can reconstruct from out-of-distribution low-fidelity data during inference, by treating them as intermediate samples in the denoising stage.

**Motivation**: one fundamental drawback in existing studies is that they assume low-fidelity data is artificially downsampled from high-fidelity data at the same timestamp. Such data inherently has more information compared to solver-generated low-fidelity data in reality, where coarser discretization grids are used in numerical solvers for saving computational resources. As illustrated in Figure 1, the former follows "integrate then downsample", which starts from the high-fidelity initial states to rollout trajectories, and then downsample at each timestamp. The latter follows "downsample then integrate", which downsamples the high-fidelity initial state to obtain coarser discretization grids

as starting points and rollout trajectories through numerical solvers. Therefore, models trained in the downsampled settings can lead to inferior performance during inference when given the solver-generated data in reality, as solver-generated low-fidelity data has large deviations/distribution shifts from high-fidelity data. This is illustrated by the visualization of low-fidelity data generated from the two: the downsampled one contains more fine-grained details than the solver-based one.

To address this, we study reconstructing high-fidelity CFD data from solver-generated low-fidelity data with initial coarser discretization grids (Sarkar et al., 2023; Ogoke et al., 2024). **Problem**: Our experiments reveal that state-of-the-art models struggle to recover fine-grained high-fidelity details given solver-generated low-fidelity inputs, due to the large distribution shifts. In response to this challenge, we present a novel diffusion model, PG-Diff, which features an *Importance Weight* strategy during training as self-guidance to locate fine-grained high-fidelity details, and a training-free *Residual Correction* module during inference for injecting physical inductive bias. The two moduli jointly guide the diffusion model toward higher-quality reconstructions. Specifically, the *Importance Weight* mod-

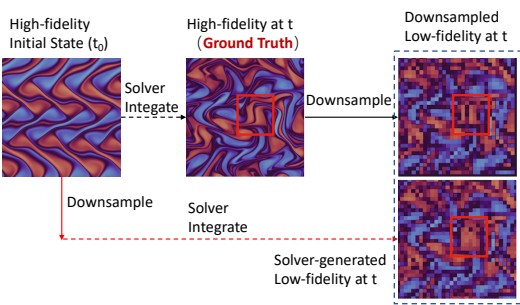

Figure 1: Comparison between downsampled (**black** line) and solver-generated (red line) flow fields. Solver-generated low-fidelity data retain less information, especially for fined-grained high-fidelity details.

ule assigns importance scores to different components in the flow field through Discrete Wavelet Transformation (DWT) (Daubechies, 1992a), which are integrated into the loss function for guiding the diffusion model to better reconstruct detailed and accurate structures. The *Residual Correction* module projects the reconstructed samples onto the solution subspace of the data governing equations, ensuring that the generated outputs also adhere to physical constraints such as Navier-Stokes Equations (Navier, 1823). We achieve this by applying gradient descent of the residuals of the governing equations at certain diffusion steps, to refine the generated high-fidelity data. We also explore different scheduling strategies on which diffusion steps to apply such correction. Our findings suggest that applying residual corrections at both the beginning and the end of diffusion steps strikes the ideal balance between reconstruction L2 error and the physical coherence measured by PDE residual.

Our key contributions are summarized as follows:

- We study a novel problem on reconstructing high-fidelity flow fields with solver-generated low-fidelity data, benefiting real-world applications. Our experiments reveal that state-of-the-art reconstruction models fail to generate high-quality outputs due to loss of fine-grained detail in low-fidelity data.

- We propose PG-Diff, a novel diffusion model for reconstructing high-quality outputs through the guidance of an *Importance Weight* strategy during training as self-guidance and a training-free *Residual Correction* method during inference as physical inductive bias.

- We present empirical evidence of PG-Diff's state-of-the-art performance in a variety of 2D turbulent flow over 4 datasets. It yields a significant improvement in terms of predictive accuracy physical consistency, and perceptual quality.

## 2 PRELIMINARIES AND RELATED WORK

We consider a machine learning model $f_\theta : \mathcal{X} \to \mathcal{Y}$ with parameters $\theta$, which transforms a data sample from low-fidelity domain $x \in \mathcal{X} \in \mathbb{R}^{m \times m}$ to high-fidelity domain $y \in \mathcal{Y} \in \mathbb{R}^{n \times n} (m < n)$. The distributions of the training and test sets for low-fidelity data are denoted by $p_\mathcal{X}^{\text{train}}$ and $p_\mathcal{X}^{\text{test}}$, respectively, and for high-fidelity data as $p_\mathcal{Y}^{\text{train}}$ and $p_\mathcal{Y}^{\text{test}}$. The training and testing distribution for low and high-fidelity data are not necessarily identical. The objective is to develop $f_\theta$ such that

it can effectively map samples from $\mathcal{X}^{\text{test}}$ to their corresponding high-fidelity counterparts $\mathcal{Y}^{\text{test}}$[1]. As diffusion model operates on the same input and output grid, we upsample the low-fidelity data uniformly to the same resolution as the target.

## 2.1 AI FOR COMPUTATIONAL FLUID DYNAMICS (CFD)

Recent advances in machine learning have led to various learning-based surrogate models for accelerating scientific discoveries (Sanchez-Gonzalez et al., 2020; Li et al., 2020; 2021). In the field of high-fidelity CFD reconstruction, researchers have developed powerful models rooted from image super-resolution domain in computer vision. Specifically, GANs (Ledig et al., 2016; Wang et al., 2019) and normalizing flows (Lugmayr et al., 2020) have achieved impressive results for image reconstruction. Later on, diffusion models (Saharia et al., 2021) have challenged the long-standing dominance of GANs. Various physical inductive biases (Raissi et al., 2019; Bai et al., 2020; Chen et al., 2021) have been injected into these methods for CFD reconstructions, which enhances the robustness and accuracy of the reconstruction task. For example, (Erichson et al., 2020; Pradhan & Duraisamy, 2021) developed super-resolution models based on Multi-Layer Perceptrons (MLPs). Fukami et al. (2019) introduced a CNN-based hybrid Downsampled Skip-Connection Multi-Scale (DSC/MS) model, while Fukami et al. (2021) adapted existing CNN models for use with moving sensors. Li & McComb (2022) further advanced the field by proposing physics-informed GANs for super-resolving multiphase fluid simulations. Additionally, Fu et al. (2023) proposed a porposal and refinement network to address the issue with limited high-fidelity data. Ren et al. (2023); Jiang et al. (2020) also leveraged CNN for spatial-temporal super resolution. These models rely on low- and high-fidelity data pairs during training and can only capture specific relationships between these pairs. As a result, when the test data deviates significantly from the training data, their performance degrades. To overcome this limitation, Gao et al. (2021) utilizes physical properties of the fluid such as conservation laws and boundary conditions for super resolution. Shu et al. (2023) leveraged diffusion model, which is trained exclusively on high-fidelity data and enables reconstruction from any type of low-fidelity input. However, these methods assume the low-fidelity data is artificially downsampled from high-fidelity sources, which limits their performance during inference when reconstructing from solver-generated low-fidelity data in reality.

## 2.2 DENOISING DIFFUSION PROBABILISTIC MODEL

Diffusion models have become a prominent class of deep generative models, demonstrating state-of-the-art performance across various domains such as image generation (Meng et al., 2022; Saharia et al., 2021; Ho et al., 2020), video synthesis (Yang & Hong, 2022; Yu et al., 2022), 3D shape creation (Zhou et al., 2021; Zeng et al., 2022), and applications in scientific fields (Shu et al., 2023; Yang & Sommer, 2023; Qiu et al., 2024). Denoising Diffusion Probabilistic Models (DDPMs) are grounded in a stochastic diffusion process, akin to those found in thermodynamics. It contains a forward and reverse process, where a data sample is gradually corrupted with noise via a Markov chain, and a neural network is trained to reverse this, progressively removing the noise. To generate new samples, a fully noisy input is denoised by the model step by step.

Formally, in the forward diffusion process, the input data $x_0$ is gradually corrupted with Gaussian noises as $q(x_t|x_{t-1}) = \mathcal{N}(x_t; \sqrt{1 - \beta_t}x_{t-1}, \beta_t \boldsymbol{I})(t = 1, 2, \cdots T)$, where $\beta_1, \ldots, \beta_T$ are fixed variance schedule that control the amount of noise introduced at each step, $T$ is the number of total diffusion steps and $q(x_t|x_{t-1})$ is the Markov transition probability. Let $\bar{\alpha}_t := \prod_{s=1}^{t} 1 - \beta_s$, we have $x_t = \sqrt{\bar{\alpha}_t}x_0 + \sqrt{1 - \bar{\alpha}_t}\epsilon_t$, which describes how to generate noisy states $x_t$ from input $x_0$. The reverse process is defined as $p_\theta(x_{t-1}|x_t) = \mathcal{N}(x_{t-1}; \mu_\theta(x_t, t), \Sigma_\theta(x_t, t))$, where $\mu_\theta$ and $\Sigma_\theta$ are learned by neural networks parameterized by $\theta$, and $p_\theta(x_{t-1}|x_t)$ is the denoising transition probability, undoing the transformation in the forward process.

Training DDPM involves minimizing a variational bound on the negative log-likelihood of $q(x_0)$. Ho et al. (2020) showed that it can be simplified as predicting the noise at each step:

$$L^{\text{simple}} = \mathbb{E}_{t,x_0,\epsilon} \left[ \|\epsilon_t - \epsilon_\theta(\sqrt{\bar{\alpha}_t}x_0 + \sqrt{1 - \bar{\alpha}_t}\epsilon_t, t)\|^2 \right], \tag{1}$$

---

[1]For direct mapping models, the training takes low- and high-fidelity pairs as inputs. For diffusion models, the training only requires high-fidelity data, and low-fidelity data is used as input during testing.

where $\epsilon_\theta(\cdot)$ is the predicted noise from DDPM's denoiser neural network. The denoiser takes noised sample $x_t = \sqrt{\bar{\alpha}_t}x_0 + \sqrt{1-\bar{\alpha}_t}\epsilon_t$ and diffusion timestamp $t$ as input to predict noise at each timestamp. $\epsilon_t \sim \mathcal{N}(\mathbf{0}, \mathbf{I})$ is the standard Gaussian noise sampled at time $t$. In this paper, we implement the backbone diffusion model proposed by Ho et al. (2020) and apply accelerated sampling techniques introduced by Song et al. (2020).

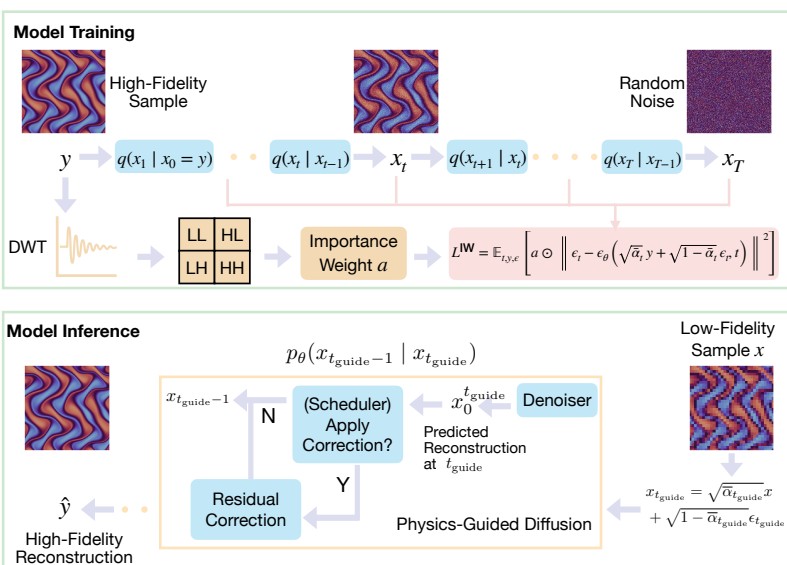

Figure 2: Training and inference pipeline of PG-Diff. Training with high-fidelity data only, guided by importance weight strategy to locate fine-grained high-fidelity details. During inference, low-fidelity data is used for reconstruction, and residual correction is applied at intermediate diffusion steps to improve physical coherence. We follow Ho et al. (2020) and use U-net as our denoiser.

# 3   METHOD:PG-DIFF

We present a novel diffusion framework PG-Diff for reconstructing high-fidelity CFD from solver-generated low-fidelity input, which has larger distribution shifts and more information loss compared to artificially downsampled low-fidelity inputs. PG-Diff features an *Importance Weight* strategy that scores different components in the flow fields through the loss function as self-guidance, forcing the model to recover more fine-grained high-fidelity details during training. In addition, a training-free *Residual Correction* module applies physics-informed correction during inference, ensuring physical coherence in reconstructed samples. The two moduli jointly guide the model towards high-quality reconstruction from wide-range of low-fidelity inputs. The overall framework is depicted in Figure 2. We now introduce each component in detail.

**Model Setup.** PG-Diff follows the guided data synthesis setting for CFD reconstruction as in (Shu et al., 2023): we train the model via recovering high-fidelity sources only, and condition on low-fidelity inputs as intermediate diffusion step during inference. This allows the model to 1.) exert control over the data generation process (reverse diffusion) during inference. Instead of starting from random noises, starting from low-fidelity inputs as intermediate diffusion steps; and 2.) reconstruct from any form of low-fidelity data, as the training does not depend on low- and high-fidelity pairs.

Formally, during the training forward process, we obtain intermediate diffusion states via the following, where $y \sim p_\mathcal{y}^{\text{train}} := x_0$ is the high-fidelity CFD that we want to recover. Here $\epsilon_t$ is sampled from a standard Gaussian distribution. $\bar{\alpha}_t := \prod_{s=1}^{t} 1 - \beta_s$ as introduced in Sec 2.2.

$$x_t = \sqrt{\bar{\alpha}_t}x_0 + \sqrt{1-\bar{\alpha}_t}\epsilon_t = \sqrt{\bar{\alpha}_t}y + \sqrt{1-\bar{\alpha}_t}\epsilon_t, \quad t = 1, 2, \cdots T. \tag{2}$$

During the inference stage, we first obtain the intermediate diffusion state $x_{t_{\text{guide}}}(0 < t_{\text{guide}} < T)$ by sequentially adding noises to the conditioned low-fidelity data $x \sim p_{\mathcal{X}}^{\text{test}}$ as

$$x_{t_{\text{guide}}} = \sqrt{\overline{\alpha}_{t_{\text{guide}}}}x + \sqrt{1 - \overline{\alpha}_{t_{\text{guide}}}}\epsilon_{t_{\text{guide}}}. \tag{3}$$

We then use $x_{t_{\text{guide}}}$ as the starting point for the reverse process to reconstruct the high-fidelity sources. Equivalently, this means the reverse diffusion starts from $t_{\text{guide}}$ instead of $T$. The reverse diffusion process progressively produces a refined high-fidelity reconstruction that aligns with the low-fidelity conditioning data, using the following DDIM (Song et al., 2020) sampling formula for acceleration.

$$x_{t-1} = \sqrt{\overline{\alpha}_{t-1}}x_0^t + \sqrt{1 - \overline{\alpha}_{t-1} - \sigma_t^2}\epsilon_\theta(x_t, t) + \sigma_t^2\epsilon_t, \quad 0 < t < t_{\text{guide}} \tag{4}$$

where $x_0^t = \frac{x_t - \sqrt{1 - \overline{\alpha}_t}\epsilon_\theta(x_t, t)}{\sqrt{\overline{\alpha}_t}}$ is the predicted reconstruction at each reverse diffusion step $t$. We use U-net as in Ho et al. (2020) to serve as our denosier, detailed in Appendix C.2.

## 3.1 IMPORTANCE WEIGHT DURING TRAINING

Existing diffusion models can fail to capture high-fidelity fine-grained details when conditioned on solver-generated low-fidelity data. We therefore introduce an importance weighting mechanism during training as self-guidance to address this limitation. Specifically, we transform the high-fidelity data into the wavelet domain using the DWT and assign importance scores to different components of fluid fields within the diffusion loss function. DWT has great abilities to locate information both in spatial and frequency domains (Daubechies, 1992b; Akansu & Haddad, 1992), thus allowing the model to capture fine-grained structures more effectively. Formally, we decompose fluid fields $y \in \mathcal{R}^{n \times n}$ into frequency subdomains, and compute the sum of squares of the high-frequency modes – namely HL (high-low), LH (low-high) and HH (high-high) subdomains as below, where $HL, LH, HH, F \in \mathbb{R}^{\frac{n}{2} \times \frac{n}{2}}$. HL captures horizontal high-frequency signals, while LH captures vertical high-frequency signals. HH captures high-frequency signals in both directions, corresponding to diagonal details such as intersections. Details about DWT are described in Appendix C.1.

$$F = HL^2 + LH^2 + HH^2. \tag{5}$$

To compute importance scores, we then uniformly upsample $F$ to $\hat{F} \in \mathbb{R}^{n \times n}$ and linearly map $\hat{F}_{i,j}$ to an importance weight $a_{i,j}$ as below, where $\alpha, \beta$ are the minimum and maximum importance weight value respectively, $Q_\theta(\hat{F}) \in \mathbb{R}$ is the $\theta$ quantile of all $\hat{F}$ values.

$$a_{i,j} = \begin{cases} \alpha + (\beta - \alpha)\frac{\hat{F}_{i,j} - Q_\theta(\hat{F})}{\max \hat{F} - Q_\theta(\hat{F})} & \text{if } \hat{F}_{i,j} > Q_\theta(\hat{F}) \\ 1 & \text{otherwise} \end{cases} \tag{6}$$

If $\hat{F}_{i,j}$ exceeds the $\theta$ quantile of all $\hat{F}$ values, the corresponding component is considered high-frequency, and will be assigned with weight greater than 1. Finally, the diffusion loss function in Eqn 1 is updated to incorporate the importance weighting as follows:

$$L^{\text{IW}} = \mathbb{E}_{t,y,\epsilon}\left[a \odot \left\|\epsilon_t - \epsilon_\theta\left(\sqrt{\overline{\alpha}_t}\,y + \sqrt{1 - \overline{\alpha}_t}\,\epsilon_t, t\right)\right\|^2\right], \tag{7}$$

where $\epsilon_\theta$ represents the predicted noise by the denoiser, $\epsilon_t$ is the ground truth noise at time $t$, and $\odot$ denotes element-wise multiplication.

**Importance Weight Design Choice.** Our *importance weight* strategy is incorporated exclusively during training, ensuring that the model focuses on regions with high-frequency details. The DWT-based calculation avoids the large computational complexity often associated with attention mechanisms. It efficiently emphasizes important features by leveraging the intrinsic properties of the wavelet transform, resulting in a more targeted learning process.

## 3.2 RESIDUAL CORRECTION DURING INFERENCE

In addition, we introduce a physics-informed residual correction module to enhance the physical coherence of the reconstructed data during inference. In the reverse diffusion process, we apply such correction to refine the reconstructed high-fidelity data at certain diffusion steps $t$ determined by a scheduling policy. Since we use DDIM (Song et al., 2020) for acceleration, we represent the sampled

diffusion states as $x^\tau$, and its corresponding reconstruction as $x_0^\tau (\tau < t_{\text{guide}})$ in the following. Note that we do not apply correction on the noisy state $x^\tau (\tau < t_{\text{guide}})$ directly. The correction is to perform gradient descent based on the residuals of the governing PDEs, detailed in Appendix D.1. The key parameters in this process are the residual correction schedule policy (the specific diffusion steps at which the correction is applied), the number of gradient descent steps $N$, and the step size $\eta$, which we study their impacts in Exp 4.5. Our findings suggest that applying residual corrections at both the beginning and the end of diffusion steps strikes the ideal balance between reconstruction L2 error and the physical coherence. We utilize the Adam algorithm Kingma & Ba (2015) for the gradient descent and summarize the physics-guided inference procedure using DDIM (Song et al., 2020) in Algorithm 1.

One existing work proposes a conditional diffusion model (Shu et al., 2023), where residual at every training diffusion step is concatenated with the current state to generate the diffusion state at next timestamp, serving as the condition. One key advantage of PG-Diff is that it is training-free, allowing the correction process to be applied without additional learning to the original diffusion model. Also, our approach offers the flexibility to adjust the residual correction schedule dynamically. This adaptability enables an optimal balance between predictive accuracy and physical consistency of the reconstructed sample. We empirically show our method achieves better performance in Sec 4.3. Additionally, diffusion model guided generation has been widely studied in Chung et al. (2023); Huang et al. (2024); Zhu et al. (2023); Shysheya et al. (2024), our Residual Correction differs by applying guidance to denoised samples instead of noisy samples. Our physical guidance also involves multiple steps of Adam gradient descent at selected backward diffusion steps, in contrast to a single step of gradient descent applied at every backward diffusion step in other works.

---

**Algorithm 1** Physics-informed Training-free Correction with DDIM during Inference.

---

**Require:** $x \in \mathcal{X}^{\text{test}}$ (guide), $t_{\text{guide}}$ ($t_{\text{guide}} < T$), $\tau = \{\tau_0, \tau_1, ..., \tau_K\}$ (sampled diffusion timestamp sequence, where $\tau_0 = 0, \cdots \tau_K = t_{\text{guide}}$. $\epsilon_\theta$ (a trained DDPM model), $\mathcal{R}(\cdot)$ (residual of the governing PDE), $N$ (number of gradient descent steps), $\eta$ (step size).

1:   $\epsilon_{\tau_K} \sim \mathcal{N}(0, I)$                                        # $\tau_K = t_{\text{guide}}$
2:   $x_{\tau_K} = \sqrt{\bar{\alpha}_{\tau_K}} x + \sqrt{1 - \bar{\alpha}_{\tau_K}} \epsilon_{\tau_K}$     # Adding noises to low-fidelity input via forward diffusion
3: **for** $i = K, K-1, \ldots, 1, 0$ **do**            # Reverse diffusion from $\tau_K$ ($\tau_K = t_{\text{guide}}$)
4:      $x_0^{\tau_i} = \frac{x_{\tau_i} - \sqrt{1 - \bar{\alpha}_{\tau_i}} \epsilon_\theta (x_{\tau_i}, \tau_i)}{\sqrt{\bar{\alpha}_{\tau_i}}}$           # Predicted reconstruction ($x_0$) at $\tau_i$
5:      **if** correction is performed at time $\tau_{i-1}$ **then**
6:          # Residual-Based Correction
7:          **repeat**
8:              $x_0^{\tau_i} = x_0^{\tau_i} - \eta \cdot \text{Adam}(\nabla \mathcal{R}(x_0^{\tau_i}))$
9:          **until** $N$ times
10:     **end if**
11:      $x_{\tau_{i-1}} = \sqrt{\bar{\alpha}_{\tau_{i-1}}} x_0^{\tau_i} + \sqrt{1 - \bar{\alpha}_{\tau_{i-1}} - \sigma_{\tau_i}^2} \epsilon_\theta (x_{\tau_i}, \tau_i) + \sigma_{\tau_i}^2 \epsilon_{\tau_i}.$     # $x_{\tau_{i-1}} \sim p_\theta(x_{\tau_{i-1}} | x_{\tau_i})$
12: **end for**
13: **return** $y = x_0$

---

## 4 EXPERIMENTS

### 4.1 DATASET

We generated four 2D turbulent flow datasets with different characteristics: 1.) *Taylor Green Vortex*, featuring how vortices diminish in turbulent flows, where large-scale vortices gradually break down into smaller turbulent structures; 2.) *Decaying Turbulence*, describing turbulence that evolves naturally without external forces. As time progresses, the turbulence weakens due to viscous effects; 3.) *Kolmogorov Flow*, which portrays turbulence influenced by a sinusoidal external force combined with a drag component; 4.) *McWilliams Flow* (Mcwilliams, 1984), which describes the behavior of isolated vortices in turbulent conditions. It is the most challenging one, demanding accurate modeling of inverse energy transfer and multi-scale vortex interactions.

The four datasets are generated by the incompressible Navier-Stokes equations following (Li et al., 2021; Kochkov et al., 2021)

$$\frac{\partial \omega(\boldsymbol{x}, t)}{\partial t} + \boldsymbol{u}(\boldsymbol{x}, t) \cdot \nabla \omega(\boldsymbol{x}, t) = \frac{1}{Re} \nabla^2 \omega(\boldsymbol{x}, t) + f(\boldsymbol{x}),$$
$$\nabla \cdot \boldsymbol{u}(\boldsymbol{x}, t) = 0, \quad \omega(\boldsymbol{x}, 0) = \omega_0(\boldsymbol{x}),$$

where $\omega$ represents the vorticity, $\boldsymbol{u}$ denotes the velocity field, $Re$ is the Reynolds number, and $f(\boldsymbol{x})$ is an external forcing term. $\omega_0$ represents the initial vorticity distribution. The PDE is numerically solved by pseudo-spectral solver (Orszag, 1972) on equispaced discretization grids. The high-fidelity data are generated with $2048 \times 2048$ discretization grid and then uniformly downsampled to $256 \times 256$, while those on the lower-resolution grids are considered low-fidelity. For each dataset, we use $80\%$ of the trajectories for training. $10\%$ for validation, and $10\%$ for testing. More details can be found in Appendix D. Note that PG-Diff also utilizes the PDE for residual correction discussed in Sec 3.2.

## 4.2 EXPERIMENT SETTINGS

**Task Setup and Baselines.** We evaluate two reconstruction settings with different low-fidelity resolutions: $64 \times 64 \rightarrow 256 \times 256$ ($4\times$ upsampling) and $32 \times 32 \rightarrow 256 \times 256$ ($8\times$ upsampling). To benchmark our approach, we compared against both direct mapping models and diffusion models: bicubic interpolation (Gonzalez & Woods, 2007), a CNN-based model (Fukami et al., 2019), a GAN-based model (Li & McComb, 2022), the vanilla diffusion model (Diff) and its conditional variant (Cond Diff) from Shu et al. (2023). In addition, we perform two ablation studies, namely PG-Diff w/o IW and PG-Diff w/o Cor , where we remove *Importance Weight* and *Residual Correction* respectively. More implementation details are provided in Appendix C.2.

**Evaluation Metrics.** We assess the reconstructed flow fields using three key metrics. We first use two standard metrics suggested by Shu et al. (2023): L2 norm for measuring the pointwise error between prediction and ground truth; unnormalized residuals of the governing equation (Res.) for assessing adherence to the underlying physics. In addition, we conduct a novel multi-scale evaluation using DWT: we transform the predicted and ground truth flow fields into wavelet space and decompose them into four subdomains: LL (low-low), LH (low-high), HL (high-low), and HH (high-high). The LL subdomain captures large-scale, low-frequency information, while LH, HL, and HH encompass higher-frequency details like turbulent structures. By calculating the L2 norm in each subdomain, we gain a comprehensive understanding of the model's performance across different scales, ensuring accurate reconstruction of both global flow features and fine-scale details.

## 4.3 RECONSTRUCTION RESULTS

Table 1 reports the mean and standard deviation of L2 and PDE residual across datasets and models. For both upsampling scales ($32 \times 32 \rightarrow 256 \times 256$ and $64 \times 64 \rightarrow 256 \times 256$), PG-Diff consistently outperforms baselines, showing its effectiveness. Notably, it achieves $3.5\%$ to $7.7\%$ performance gain against baselines in the $4\times$ upsampling setting. The lower L2 achieved by PG-Diff indicates that PG-Diffeffectively captures the essential features and dynamics of the turbulent flows, while the significantly reduced PDE residuals demonstrate that PG-Diff's predictions adhere more closely to the underlying physical laws governing fluid dynamics. Reconstructions from direct mapping models (GAN, CNN) exhibit significant deviations in physical coherence. For complex datasets dominated by fine-grained details, such as *Kolmogorov Flow* and *McWilliams Flow*, PG-Diff consistently outperforms baselines by a big margin, showing better reconstruction accuracy and physical coherence. Through ablation studies, we demonstrate that both *Importance Weight* and *Residual Correction* contribute significantly in improving model's performance, underscoring the effectiveness of our design choices in capturing the complex behaviors of turbulent flows.

**Multi-Scale Evaluation.** In addition, we assess the model's ability to capture flow structures at different scales using DWT. As shown in Figure 12, PG-Diff demonstrates superior performance in the LL, LH, and HL subdomains, achieving the best or near-best results among all methods. Excelling in the LL subdomain indicates a strong capability in capturing large-scale, low-frequency components of the turbulent flows. The superior performance in the LH and HL subdomains suggests the effectiveness of PG-Diff in capturing small-scale vortices and transitions between scales. While

| | | Bicubic* | CNN | GAN | Diff | Cond Diff | PG-Diff | PG-Diff w/o Cor | PG-Diff w/o IW |
|---|---|---|---|---|---|---|---|---|---|
| *Taylor Green Vortex* | L2 | 6.08 | **3.10±0.07** | 3.36±0.08 | 3.28±0.01 | 4.50±0.04 | 3.18±0.01 | 3.20±1e-4 | 3.22±0.01 |
| | Res. | 2.18e4 | 1.40e5±3.6e4 | 4.54e4±1.3e4 | 4.17e4±3.3e3 | **1.36e3±42** | 5.31e3±40 | 7.44e4±1.4e4 | 6.95e3±569 |
| | L2 | 3.04 | 2.99±0.06 | 2.83±0.3 | 1.68±2e-3 | 3.57±0.02 | **1.55±9e-3** | 1.57±0.02 | 1.59±6e-3 |
| | Res. | 1.05e4 | 1.37e5±3.6e4 | 4.18e4±1.1e4 | 3.06e4±4.3e3 | 1.47e5±1.9e4 | **4.39e3±3** | 6.67e4±8.4e3 | 6.09e3±9 |
| *Decaying Turbulence* | L2 | 3.62 | 1.81±0.1 | 2.09±0.02 | 1.84±1e-3 | 2.02±8e-4 | **1.78±1e-3** | 1.78±2e-3 | 1.79±3e-3 |
| | Res. | 3.41e3 | 3.05e3±500 | 7.63e2±111 | 9.97e3±460 | 3.49e4±1.7e4 | **1.65e2±11** | 9.86e2±165 | 2.84e2±22 |
| | L2 | 1.74 | 1.66±0.1 | 2.06±0.05 | 0.85±6e-4 | 1.34±7e-4 | **0.82±1e-3** | 0.83±6e-3 | 0.83±4e-3 |
| | Res. | 1.31e3 | 3.16e3±781 | 6.65e2±59 | 4.33e3±164 | 1.52e4±161 | **85±0.9** | 2.76e3±1.5e3 | 1.87e2±37 |
| *Kolmogorov Flow* | L2 | 4.71 | 3.02±8e-3 | 3.14±0.04 | 3.09±1e-3 | 3.13±1e-3 | **2.78±8e-3** | 2.82±1e-3 | 2.93±5e-4 |
| | Res. | 2.63e3 | 3.52e3±125 | 3.30e2±87 | 1.80e2±22 | 80±2 | **40.12±0.3** | 6.68e2±10 | 1.40e2±83 |
| | L2 | 3.22 | 2.22±0.01 | 3.02±0.04 | 1.79±1e-3 | 1.79±1e-3 | **1.66±4e-3** | 1.69±8e-4 | 1.73±1e-3 |
| | Res. | 1.86e4 | 7.30e2±49 | 3.03e2±92 | 3.09e2±29 | 1.65e2±7 | **39±0.06** | 9.27e2±1 | 3.33e2±145 |
| *McWilliams Flow* | L2 | 3.19 | **2.00±0.07** | 2.03±0.05 | 2.23±4e-3 | 2.24±5e-4 | 2.04±2e-4 | 2.06±3e-5 | 2.16±3e-4 |
| | Res. | 3.35e2 | 6.50e2±180 | 2.86e2±51 | 16±0.9 | 12±1 | **5.5±4e-3** | 66±1 | 7.8±0.2 |
| | L2 | 2.17 | 1.75±0.1 | 1.96±0.08 | 1.29±3e-4 | 1.30±4e-4 | **1.24±9e-5** | 1.27±1e-4 | 1.30±2e-4 |
| | Res. | 2.17e2 | 3.64e2±127 | 2.42e2±35 | 21±4 | 31±4 | **6.36±2e-3** | 88±0.7 | 12±0.06 |

Table 1: Quantitative performance comparison over four datasets on L2 and Res. Results are repeated three times. Metrics are reported for both 32×32 → 256×256 ( grey ) and 64×64 → 256×256 tasks. **Bold** values indicate the best performance and underlined values represent the second-best. * Bicubic interpolation is a deterministic algorithm, which has zero standard deviation.

direct mapping models (CNN, GAN) exhibit better performance in the HH subdomain, PG-Diff's results are close to those best-performing methods. The balanced performance across all subdomains demonstrates that PG-Diff can reconstruct both large-scale structures and fine-grained details essential for turbulent flows. Additional details on other datasets are presented in the Appendix E.2.

**Runtime Comparison.** We report the runtime comparison of different methods in Appendix E.8. PG-Diff only increases small inference time while achieving superior performance improvement against baselines. The total runtime of using the numerical solvers to generate low-fidelity data, and then reconstructing with PG-Diff is considerably faster than the time required to produce samples with similar error, demonstrating the effectiveness of PG-Diff.

**Sensitivity Analysis.** We study the effects of three key hyperparameters in calculating the importance weight in Sec 3.1: the maximum importance weight $\beta$, minimum importance weight $\alpha$, and the threshold parameter $\theta$. The results in Appendix E.5 show that increasing $\beta$ reduces both L2 and PDE residuals, indicating a broader range of importance weights is beneficial. Smaller $\alpha$ values improve performance, while setting $\alpha$ to 1 is suboptimal. This can be understood as it decreases the difference between high-frequency and low-frequency regions, as the weight for the latter one is set to 1. Additionally, a larger $\theta$ (0.7 or 0.8) helps the model focus on important details.

## 4.4 VISUALIZATIONS

We visualize the reconstructed high-fidelity data conditioned on low-fidelity inputs in Figure 3. PG-Diff consistently captures more fine-grained details, resulting in a closer resemblance to the high-fidelity ground truth compared to other methods. This is particularly evident in regions with complex vortical structures and turbulent features, where competing models often smooth out finer details. An example is the reconstructions from CNN and GAN on the *McWilliams FLow* dataset. While direct mapping methods can achieve a low L2 norm, their reconstructed samples often merge small fine-grained regions into one large blurred region, leading to information loss and significant divergence from the original high-fidelity data. By comparison, PG-Diff is able to recover the sharp edges, turbulence, and subtle variations in the flow field. We additionally show the strong reconstruction ability of our model variants compared to baselines in Appendix E.3. We also use the Learned Perceptual Image Patch Similarity (LPIPS) score (Zhang et al., 2018) to measure the perceptual quality of reconstructed samples in Appendix E.4.

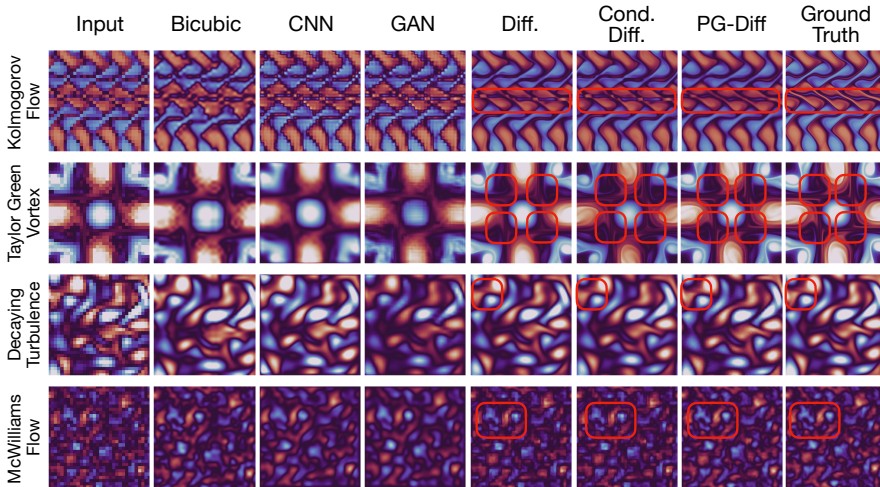

Figure 3: Visualization of reconstructed high-fidelity data across datasets. We highlight fine-grained details where PG-Diff achieves better performance than baseline diffusion models.

## 4.5 PHYSICAL GUIDANCE

We conduct a systematic study on how to schedule the residual correction during inference towards the best performance.

**Scheduling Policy.** We first explore what would be the optimal schedule policy by comparing against: 1.) *Uniform N*: Distributing N residual correction steps evenly across diffusion steps; 2.) *Start M, End N* Placing M consecutive correction steps at the start and N at the end of the diffusion process; 3.) *Start N, Space S*: Placing N correction steps at the start with a spacing of S; 4.) *End N, Space S*: Placing N correction steps at the end with a spacing of S. Results in Table 2 suggest that applying correction steps at the beginning achieves the lowest L2, while applying at the end has the lowest PDE residuals. To balance between L2 and PDE residual, we

| Schedule | $32\times32 \to 256\times256$ | | $64\times64 \to 256\times256$ | |
| --- | --- | --- | --- | --- |
| | L2 | Res. | L2 | Res. |
| Uniform 4 | 2.7910 | 26.32 | 1.6645 | 23.50 |
| Start 3 End 1 | 2.7906 | 30.00 | 1.6617 | 32.51 |
| Start 2 End 2 | 2.7897 | 40.12 | 1.6609 | 39.97 |
| Start 1 End 3 | 2.7896 | 59.79 | 1.6617 | 59.39 |
| Start 4 Space 1 | **2.7889** | 101.60 | **1.6587** | 105.84 |
| Start 4 Space 2 | **2.7888** | 98.32 | **1.6590** | 102.14 |
| Start 4 Space 3 | **2.7887** | 95.36 | **1.6594** | 95.08 |
| End 4 Space 1 | 2.7930 | **19.30** | 1.6668 | **21.97** |
| End 4 Space 2 | 2.7932 | **17.55** | 1.6747 | **20.96** |
| End 4 Space 3 | 2.7924 | **19.26** | 1.6730 | **23.57** |

Table 2: Scheduling policy comparison on *Kolmogorov Flow*. We emphasize that applying residual correction at the end leads to a reduced PDE residual, while placing it at the beginning achieves a lower L2 loss.

adopt the *Start N End N* schedule, which places $N$ correction steps at both the beginning and the end of the diffusion process. We next study the optimal number of $N$.

**Number of Correction Steps.** We vary the number of $N$ in the *Start N, End N* policy as shown in Figure 4. We observe that increasing $N$ leads to enhanced physical coherence measured by PDE residuals. However, L2 does not continuously decrease with larger $N$. It reaches a minimum when $N = 2$. This suggests that while more correction steps improve models' adherence to physical laws, an excessive number may interfere with the model's ability to accurately capture the intricate details of the turbulent flow. Therefore, we use *Start 2, End 2* as the optimal balance in our method.

## 4.6 MODEL GENERALIZATION

We observe that PG-Diff generalizes well even beyond its training distributions. Specifically, we conduct evaluations over three generalization settings: time discretization in numerical solver, spatial domain size, and Reynolds number as shown in Table 3. We train PG-Diff on the original *Kolmogorov Flow* dataset configurations, which has solver integration timestep as $dt = 1/32$, spa-

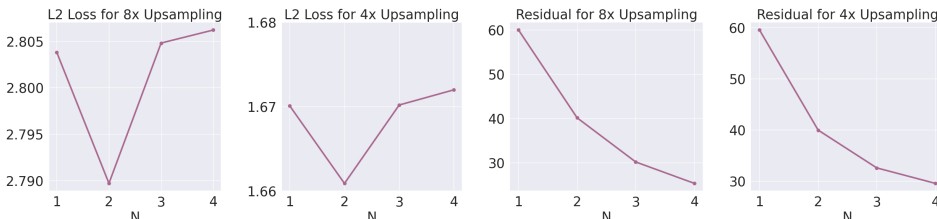

Figure 4: Varying number of $N$ using the *Start N, End N* correction schedule on the *Kolmogorov Flow* dataset. Showing both L2 and Residual across upsampling settings.

tial domain size ranging from $2\pi \times 2\pi$, and Reynolds number as $Re = 1000$. We then directly test these new configurations on the trained model, without any additional retraining or fine-tuning. We compare the performance against models directly trained one each new configuration.

The results reveal that the pertained PG-Diff performs comparably to trained ones directly on each new configuration. This underscores our model's strong generalization capabilities across different flow conditions. Such generalization ability can due to the fact that both our *importance weight* mechanism and *residual correction* modules are training-free. They enable our model to locate fine-grained high-fidelity details and adhere to physical laws independent of training data, showing their strong superiority. A similar trend is also observed in $4\times$ upsampling experiments, shown in Appendix E.7.

| Variation | Model | L2 | Res. |
|---|---|---|---|
| **Time Discretization Variations** | | | |
| $dt = 1/40$ | Trained on Original Data | 2.8144 | 34.29 |
| | Trained on $dt = 1/40$ Data | 2.8136 | 34.52 |
| $dt = 1/50$ | Trained on Original Data | 2.7761 | 37.11 |
| | Trained on $dt = 1/50$ Data | 2.7795 | 30.60 |
| **Spatial Domain Size Variations** | | | |
| $1\pi \times 1\pi$ | Trained on Original Data | 1.9045 | 262.42 |
| | Trained on $1\pi \times 1\pi$ Data | 1.7672 | 190.99 |
| $1.5\pi \times 1.5\pi$ | Trained on Original Data | 2.3573 | 80.66 |
| | Trained on $1.5\pi \times 1.5\pi$ Data | 2.3028 | 88.88 |
| **Reynolds Number Variations** | | | |
| $Re = 500$ | Trained on Original Data | 2.3694 | 24.11 |
| | Trained on $Re = 500$ Data | 2.3542 | 26.96 |
| $Re = 2000$ | Trained on Original Data | 3.2739 | 21.21 |
| | Trained on $Re = 2000$ Data | 3.2914 | 22.48 |

Table 3: Generalization results on *Kolmogorov Flow* dataset with $32 \times 32 \rightarrow 256 \times 256$ setting.

## 5 CONCLUSION

We study a novel problem of reconstructing high-fidelity CFD from solver-generated low-fidelity inputs in a practical setting across scientific domains. We present a novel diffusion model PG-Diff, which achieves high-quality reconstruction guided by the proposed *Importance Weight* and *Residual Correction* modulus jointly, yielding state-of-the-art performance across four 2D turbulent flow datasets.

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

CONTENTS

## A  LIMITATIONS

Our current research on super-resolution in computational fluid dynamics (CFD) focuses exclusively on vorticity data. However, future work can broaden its scope to include the reconstruction of high-fidelity velocity and pressure fields. These additional physical quantities are critical for capturing a more comprehensive representation of fluid dynamics.

## B  BROADER IMPACTS

By enhancing the accuracy of CFD simulations, this approach can significantly reduce the computational costs associated with high-resolution simulations, which are often prohibitively expensive in terms of both time and resources. This has implications for a wide range of industries, including aerospace, automotive design, and environmental engineering, where high-fidelity simulations are essential for optimizing performance, safety, and sustainability. Additionally, by improving the quality of low-resolution CFD data, our work could enable more accessible, efficient research and development processes, allowing smaller organizations and research teams to leverage advanced simulation capabilities.

## C  MODEL DETAILS

### C.1  DISCRETE WAVELET TRANSFORMATION

The Discrete Wavelet Transform (DWT) decomposes an image $I \in \mathbb{R}^{n \times n}$ into different frequency components by successively applying low-pass and high-pass filters. Given an image $I$, the DWT first applies a low-pass filter $g[n]$ and a high-pass filter $h[n]$ along each row, producing two sub-bands: $I_{\text{low}}[i,j] = \sum_k I[i,k] \cdot g[2j-k]$ and $I_{\text{high}}[i,j] = \sum_k I[i,k] \cdot h[2j-k]$, each downsampled by a factor of 2 along the columns, resulting in matrices of shape $n \times \frac{n}{2}$.

Next, the same filters are applied to each column of $I_{\text{low}}$ and $I_{\text{high}}$, generating four sub-bands:

$$LL[i,j] = \sum_k I_{\text{low}}[k,j] \cdot g[2i-k], \quad LH[i,j] = \sum_k I_{\text{low}}[k,j] \cdot h[2i-k],$$

$$HL[i,j] = \sum_k I_{\text{high}}[k,j] \cdot g[2i-k], \quad HH[i,j] = \sum_k I_{\text{high}}[k,j] \cdot h[2i-k].$$

The resulting sub-bands $LL$, $LH$, $HL$, and $HH$ each have dimensions $\frac{n}{2} \times \frac{n}{2}$, capturing the low-low, low-high, high-low, and high-high frequency components, respectively, of the original image.

- $LL$: This sub-band captures the approximation or low-frequency content of the image in both the horizontal and vertical directions. It retains most of the important structural information of the image.
- $LH$: This sub-band captures the vertical high-frequency components (edges and fine details) of the image. It primarily highlights vertical details like vertical edges.
- $HL$: This sub-band captures the horizontal high-frequency components of the image. It emphasizes horizontal edges or features.
- $HH$: This sub-band captures the high-frequency content in both horizontal and vertical directions, corresponding to diagonal details such as corners, intersections, and textures.

These sub-bands collectively represent the image in both spatial and frequency domains. higher values in the $LH$, $HL$, and $HH$ sub-bands correspond to regions with higher frequency variations (such as edges or fine details) in the respective subdomains. Higher values in the $LL$ sub-band correspond to regions with higher intensity or brightness in the original image but represent low-frequency content.

### C.2  IMPLEMENTATION DETAILS

We implement all models in PyTorch and show their implementation details below.

**Bicubic** Bicubic interpolation is a widely used image scaling technique that enhances the resolution of low-resolution images by using a weighted average of pixels in a 4x4 neighborhood around each pixel. Bicubic interpolation uses 16 neighboring pixels to produce smoother and more accurate results. This method is deterministic and does not involve any machine learning or training.

**CNN** We modify the hybrid Downsampled Skip-Connection Multi-Scale (DSC/MS) architecture proposed by Fukami et al. (2019). In the original implementation of the DSC/MS model, the model operates on a single frame of velocity field. We modify the model to work in our settings, where the input consists of three consecutive frames of low-fidelity vorticity data, upsampled to $256 \times 256$. The DSC component starts by compressing the input through a series of convolutional layers, effectively capturing multi-scale features. Skip connections are then introduced at various stages to preserve spatial information, ensuring that the network retains fine details from earlier layers. The MS component consists of three parallel convolutional paths that capture features at different scales and combine them to produce the high-resolution output of three consecutive frames of $256 \times 256$ vorticity data. The training process uses the L2 loss function, running for 300 epochs with a learning rate of $1e-5$.

**GAN** We modify the physics-informed GAN proposed by Li & McComb (2022). In the original implementation of the physics-informed GAN, the input was low-resolution phase fraction data from a multiphase fluid simulation, and the output was a high-resolution phase fraction representation. Our

adaptation uses low-fidelity vorticity data upsampled to $256 \times 256$ as input and high-fidelity vorticity data as output. The generator consisted of multiple convolutional layers and residual blocks, while the discriminator had a series of convolutional layers followed by dense layers. The physics-informed loss function combined a mean squared error (MSE) term with an additional term designed to enforce conservation of mass, ensuring the generated data remained consistent with fluid dynamics principles. The model was trained for 300 epochs with a learning rate of $1e - 5$.

**Diffusion** The unconditional diffusion model in Shu et al. (2023) is trained only on high-fidelity CFD data without requiring low-fidelity data during training. The model uses a U-Net architecture with hierarchical convolutional blocks, multi-level skip connections, and a self-attention mechanism in the bottleneck layer to capture complex features of the high-fidelity data. The training process spans 300 epochs, where the model learns to predict and remove noise added during the forward diffusion process. During inference, the model starts with a guided reverse diffusion process from an intermediate timestep $t < T$ instead of starting with complete random Gaussian noise. The low-fidelity input data is first upsampled to the target resolution and then combined with Gaussian noise to form the initial input for the reverse diffusion process. This noisy data serves as guidance for generating the high-fidelity output.

**Conditional Diffusion** The conditional diffusion model proposed in Shu et al. (2023) incorporates physics-informed guidance during both training and inference by utilizing the gradient of the PDE residual with respect to the noised sample. During training, at each step of the forward diffusion process, the residual of the governing PDE is computed based on the current noised sample. The gradient of this residual is calculated and concatenated with the noised sample as input to the U-Net. During inference, the same process is applied: the model computes the gradient of the PDE residual at each reverse diffusion step and concatenates it with the noised sample.

**PG-Diff** Starting with $x_t$, which represents an intermediate denoised state at time-step $t$, the denoising process utilizes a U-Net model to estimate $\epsilon_\theta$. The UNet architecture starts with an initial 3x3 convolution that maps the `in_channels:3` to `ch:128`. The encoder progressively downsamples at each level, using a channel multiplier `ch_mult:[1, 1, 1, 2]`, which means the channels stay at 128 for three levels and then double to 256. Attention blocks are applied at resolutions to capture long-range dependencies. The decoder mirrors the encoder, using upsampling layers to restore spatial resolution and skip connections to preserve details. The output is transformed back to the original `out_ch:3` channels through a final 3x3 convolution. The estimated $\epsilon_\theta$ is then integrated into a sampling process, outlined in detail by Steps 5 of Algorithm 1, to iteratively generate $x_{t-1}$. In this approach, the inference phase is governed by several critical hyperparameters: $t$, which defines where the partial backward diffusion sampling begins; $K$, the total number of DDIM backward diffusion steps performed; and the set $\{\beta_t\}_{t=1}^{T}$, which dictates the scaling factors controlling noise variance throughout the forward diffusion stages. We adopt the same hyperparameters used in Shu et al. (2023). The training algorithm for our PG-Diff is presented in Algorithm 2. The PG-Diff training procedure involves iteratively updating the denoiser $\epsilon_\theta$ using high-fidelity training data $\mathcal{Y}^{\text{train}}$. In each training iteration, a sample $y$ is drawn from the high-fidelity training dataset, and a time-step $t$ is randomly selected from a uniform distribution over $\{1, \ldots, T\}$. Gaussian noise $\epsilon_t$ is then sampled from $\mathcal{N}(0, I)$. The high-fidelity data $y$ is transformed into the wavelet domain to compute an importance weight $a$. The model's parameters $\theta$ are updated through a gradient descent step using the gradient $\nabla_\theta a \odot \|\epsilon_t - \epsilon_\theta(\sqrt{\bar{\alpha}_t} y + \sqrt{1 - \bar{\alpha}_t} \epsilon_t, t)\|^2$. This process repeats until the model converges, enabling the denoiser to learn to reconstruct fine-grained details in the high-fidelity data effectively.

# D  DATASET

All datasets are generated using pseudo-spectral solver implemented by Li et al. (2020). Our dataset will be released upon publication of our paper and be free to use.

The time-stepping method employed is a combination of the Crank-Nicholson scheme and Heun's method. The Crank-Nicholson scheme is an implicit, second-order accurate method, and it is applied to the viscous term. This allows the solver to be stable even for relatively large time steps when handling viscous diffusion. Heun's method, a second-order Runge-Kutta technique, is used to handle the non-linear advection term. The combination of these two methods provides an efficient and

---

**Algorithm 2** PG-Diff Model Training.

---

**Require:** $\mathcal{Y}^{\text{train}} \sim p_{\mathcal{Y}}^{\text{train}}$ (High-fidelity data used for training), $\epsilon_\theta$ (Denoiser to be trained)

1: **repeat**
2:    $y \in \mathcal{Y}^{\text{train}}$
3:    $t \sim \text{Uniform}(\{1, \dots, T\})$
4:    $\epsilon_t \sim \mathcal{N}(0, I)$
5:    Transform $y$ into the wavelet domain and compute the importance weight $a$
6:    Take a step of gradient descent on $\theta$ with the following gradient:

$$\nabla_\theta a \odot \|\epsilon_t - \epsilon_\theta(\sqrt{\bar{\alpha}_t}y + \sqrt{1 - \bar{\alpha}_t}\epsilon_t, t)\|^2$$

7: **until** converged

---

accurate way to evolve the vorticity field over time, with the implicit Crank-Nicholson step ensuring stability for stiff viscous terms, and Heun's method capturing the non-linearity of the advection.

For numerical stability, the solver uses an adaptive time-stepping approach governed by the Courant-Friedrichs-Lewy (CFL) condition. The CFL condition ensures that the time step remains sufficiently small relative to the velocity field and the external forcing, preventing instabilities that can arise from rapid changes in the solution. Additionally, the dealiasing procedure, using the 2/3 rule, removes high-frequency components from the Fourier spectrum, ensuring that non-physical aliasing effects are avoided.

---

**Algorithm 3** Pseudo-spectral Navier-Stokes Solver

---

1: **Input:** Initial vorticity $\omega_0$, forcing $f$, Reynolds number $Re$, total time $T$, timestep $\Delta t$, domain size $L_1$, $L_2$, grid size $s_1$, $s_2$, adaptivity flag
2: **Output:** Vorticity $\omega$ at time $T$
3: **Initialize:** Compute wavenumbers, Laplacian, and dealiasing mask
4:
5: **while** $t < T$ **do**
6:    **if** adaptive **then**
7:       Compute velocity field $\boldsymbol{u} = \nabla^\perp \psi$
8:       Update timestep $\Delta t$ based on CFL
9:    **end if**
10:    Compute non-linear term in Fourier space
11:    Predictor and corrector steps using Crank-Nicholson + Heun
12:    Apply dealiasing mask
13:    Update time $t = t + \Delta t$
14: **end while**
15: **return** Vorticity $\omega$

---

**Taylor Green Vortex** The initial vorticity field is based on the analytical solution of the TGV, and to generate different trajectories, we added random perturbations from a Gaussian random field. These perturbations introduce variability to the initial conditions while maintaining the overall vortex structure. No external forcing was applied during the simulation, and the spatial domain is $[0, \frac{3}{2}\pi]^2$ with a fixed Reynolds number of 1000. The simulation used a time step $dt = \frac{1}{32}$, and 100 trajectories were generated each with a total duration of $T = 6$ seconds. The initial vorticity field is based on the analytical solution for the two-dimensional periodic domain. The vorticity field $\omega$ is initialized as $\omega = -2U_0k\sin(kx)\sin(ky)$, where $U_0$ is the initial velocity amplitude, and $k$ is the wave number that determines the size of the vortices. To introduce variability and generate different trajectories, a Gaussian Random Field is added as a perturbation to the initial vorticity.

**Decaying Turbulence** The spatial domain is $[0, 1]^2$, with periodic boundary conditions and a fixed Reynolds number of 450. The simulation used a time step $dt = \frac{1}{32}$, and 400 trajectories were generated, each with a total duration of $T = 2$ seconds. The initial conditions for the decaying turbulence dataset are generated by superimposing randomly positioned vortices of varying intensity and size. Each vortex is characterized by a randomly selected core size and maximum rotational velocity, allowing for a diverse range of initial flow structures. The vortices are distributed randomly

throughout the domain, and their periodic images are added to ensure the proper enforcement of periodic boundary conditions.

**Kolmogorov Flow** The initial vorticity field is generated using a Gaussian random field, and the system is subjected to a forcing term of the form $f(\boldsymbol{x}) = -4\cos(4x_2) - 0.1\omega(\boldsymbol{x}, t)$. This forcing drives the flow in the y-direction while introducing a drag force that dissipates energy. The spatial domain is $[0, 2\pi]^2$, with a fixed Reynolds number of 1000. The simulation used a time step $dt = \frac{1}{32}$, and 50 trajectories were generated, each with a total duration of $T = 10$ seconds. The initial vorticity field is produced by sampling from a Gaussian Random Field. As the external forcing continually adds energy to the system, the initially simple vorticity evolves into intricate and turbulent structures. The vorticity is allowed to evolve over a 5-second period, and the final state at the end of this interval is used as the initial condition for our dataset.

**McWilliams Flow** The phenomenon illustrates the emergence of order from initially disordered turbulent motion, driven by viscous dissipation and the self-organization of the flow. No external forcing is applied during the simulation, allowing for a natural decay and evolution of the turbulence. The spatial domain is $[0, 2\pi]^2$, with periodic boundary conditions and a Reynolds number of 2000, providing a high degree of turbulence. The simulation used a time step $dt = \frac{1}{32}$, and 50 trajectories were generated, each with a total duration of $T = 10$ seconds. The initial vorticity field for the *McWilliams Flow* is generated following the method described by Mcwilliams (1984). The process begins by constructing a Fourier mesh over the spatial domain, where the wavenumbers $k_x$ and $k_y$ are calculated. A scalar wavenumber function is prescribed, and the ensemble variance is determined to ensure that the energy distribution in Fourier space follows the desired spectral shape. Random Gaussian perturbations are applied to each Fourier component of the stream function, producing a random realization of the vorticity field. To ensure the stream function has a zero mean, a spectral filter is applied, and the field is normalized based on the kinetic energy. Finally, the vorticity field is computed in physical space by taking the inverse Laplacian of the stream function in Fourier space, resulting in a turbulent flow field that evolves naturally without external forcing.

We show the visualization of the four datasets below.

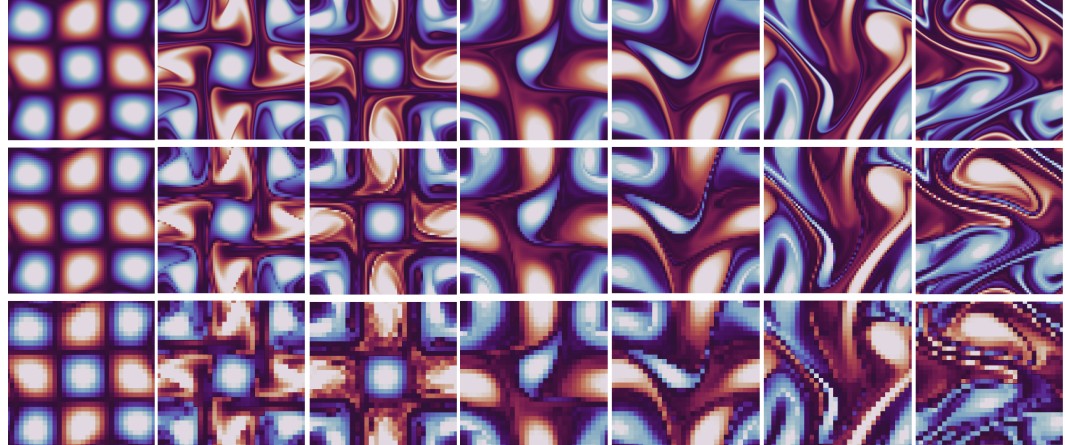

Figure 5: Example trajectory of the Taylor Green Vortex dataset. The first row is the high-fidelity data with $256 \times 256$ discretization grid. The second and third rows are low fidelity data with $64 \times 64$ and $32 \times 32$ discretization grids, respectively. For visualization, we upsample the low–fidelity data to $256 \times 256$ discretization grid.

### D.1 RESIDUAL CALCULATION

The PDE residual of the governing equation $\mathcal{R}(\omega)$ is defined as

$$\mathcal{R}(\omega) = \frac{1}{N} \sum_{i,j} \left[ \text{LHS}(\omega)_{i,j} - \text{RHS}(\omega)_{i,j} \right]^2$$

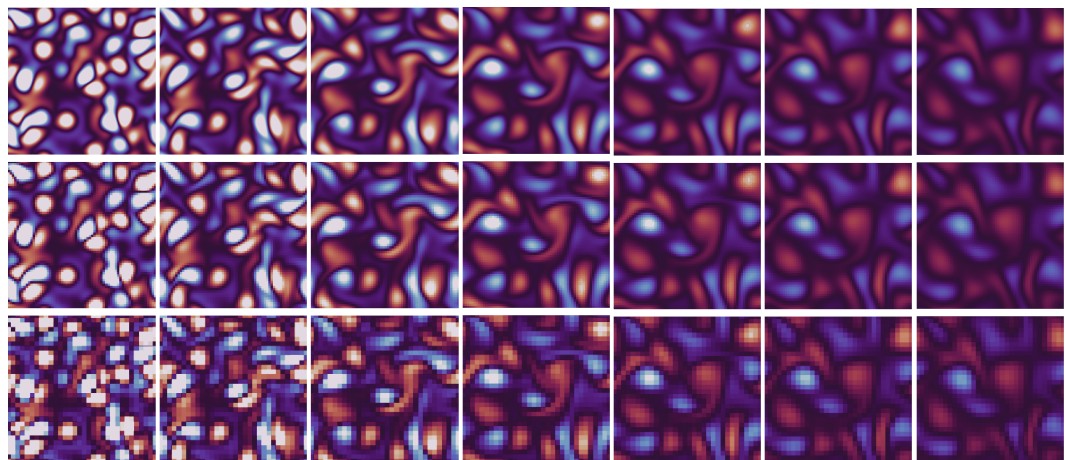

Figure 6: Example trajectory of the Decaying Turbulence dataset. The first row is the high-fidelity data with $256 \times 256$ discretization grid. The second and third rows are low fidelity data with $64 \times 64$ and $32 \times 32$ discretization grids, respectively. For visualization, we upsample the low–fidelity data to $256 \times 256$ discretization grid.

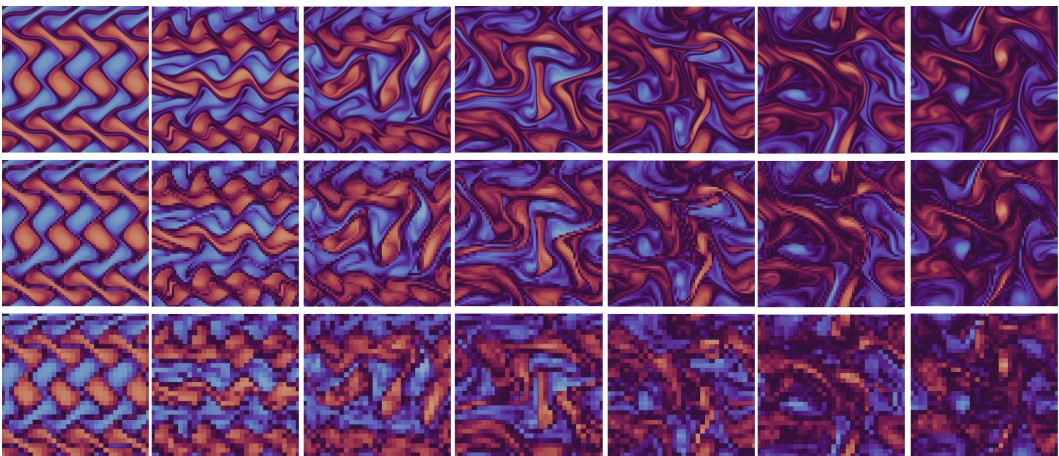

Figure 7: Example trajectory of the *Kolmogorov Flow* dataset. The first row is the high-fidelity data with $256 \times 256$ discretization grid. The second and third rows are low fidelity data with $64 \times 64$ and $32 \times 32$ discretization grids, respectively. For visualization, we upsample the low–fidelity data to $256 \times 256$ discretization grid.

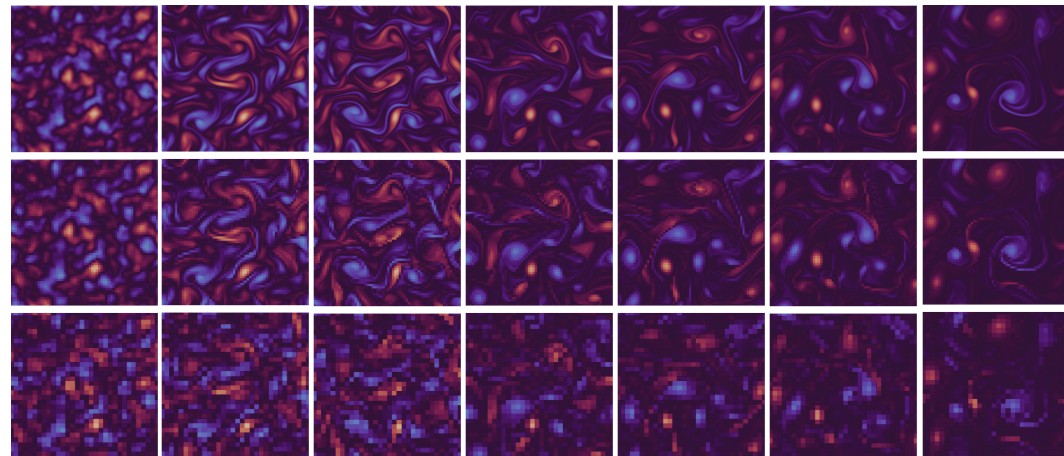

Figure 8: Example trajectory of the *McWilliams Flow* dataset. The first row is the high-fidelity data with $256 \times 256$ discretization grid. The second and third rows are low fidelity data with $64 \times 64$ and $32 \times 32$ discretization grids, respectively. For visualization, we upsample the low–fidelity data to $256 \times 256$ discretization grid.

where $\text{LHS}(\omega)$ and $\text{RHS}(\omega)$ represent the left-hand side and right-hand side expressions of the PDE, respectively. $\omega$ represents the vorticity. $N$ represents the number of grid points. For the incompressible navier stokes equation tested in this paper, the residual is defined as

$$\mathcal{R}(\omega) = \frac{1}{N} \sum_{i,j} \left[ \frac{\partial \omega(\boldsymbol{x},t)}{\partial t} + \boldsymbol{u}(\boldsymbol{x},t) \cdot \nabla \omega(\boldsymbol{x},t) - \frac{1}{Re} \nabla^2 \omega(\boldsymbol{x},t) - f(\boldsymbol{x}) \right]^2$$

# E  ADDITIONAL EXPERIMENTAL RESULTS

## E.1  DWT VISUALIZATION

Figure 9 presents sample importance masks identified as high-frequency features using the DWT. These masks highlight the regions where our model focuses to capture intricate details. The samples clearly demonstrate that our method effectively captures the fine-grained structures in the turbulent flow, validating its ability to reconstruct small-scale vortices and turbulence that are often missed by conventional approaches.

| High Fidelity Vorticity Field | Importance Mask | High Fidelity Vorticity Field | Importance Mask |

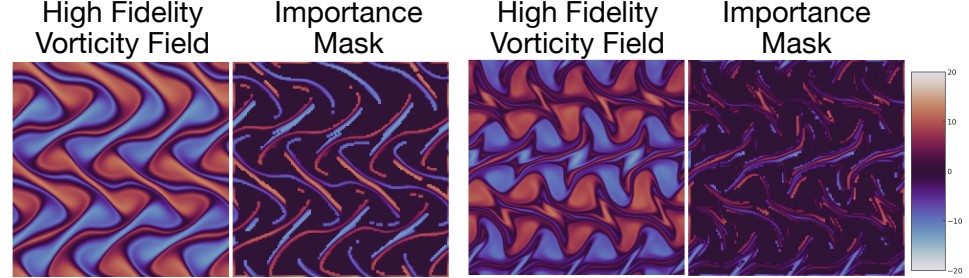

Figure 9: High frequenct components in the *Kolmogorov Flow* field identified by discrete wavelet transformation. We compare the high fidelity vorticity field with the importance mask.

## E.2  MULTI-SCALE EVALUATION

To comprehensively evaluate the model's ability to capture both overall flow dynamics and intricate details, we performed a multi-scale analysis by transforming the predicted and ground truth flow fields into the wavelet domain. This transformation produced four subdomains: LL (low-low),

LH (low-high), HL (high-low), and HH (high-high). The LL subdomain represents broad, low-frequency components, while the LH, HL, and HH subdomains capture increasingly finer, high-frequency features, including turbulent structures. We then assessed the L2 norm across these sub-domains to gain deeper insights into the model's effectiveness in reconstructing both large-scale patterns and fine-grained details.

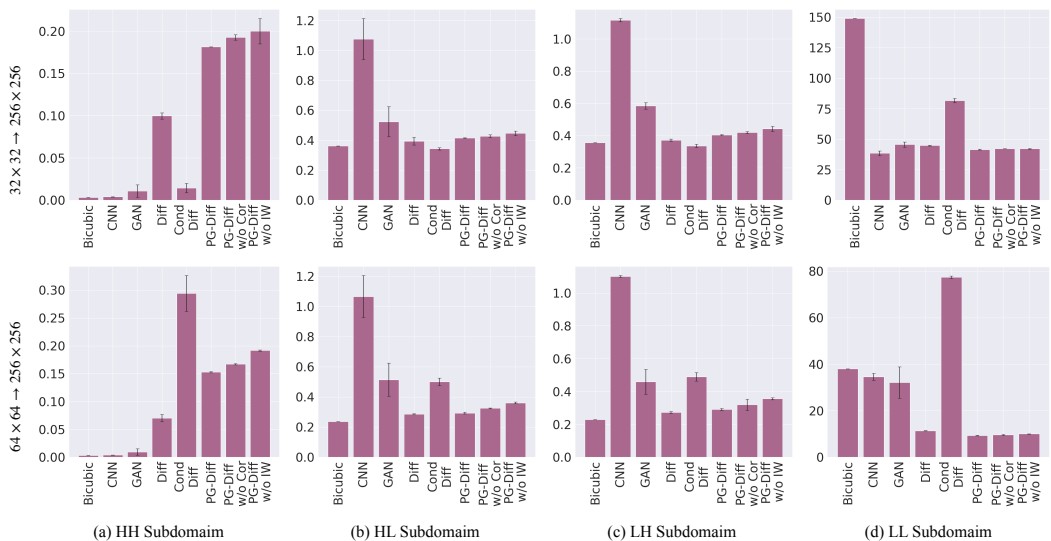

Figure 10: Wavelet subdomain L2 for the *Taylor Green Vortex* dataset.

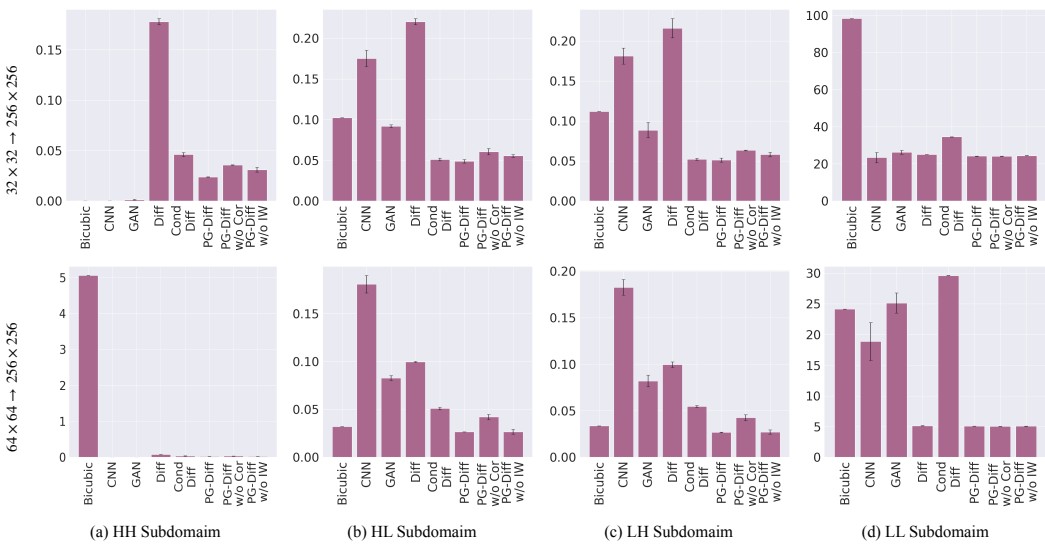

Figure 11: Wavelet subdomain L2 for the *Decaying Turbulence* dataset.

### E.3 RECONSTRUCTION VISUALIZATION

We present visualizations of the reconstructed high-fidelity data using PG-DIFF w/o Cor across different methods and datasets. As shown in Figure 14 and Figure 15, PG-Diff consistently produces impressive qualitative results.

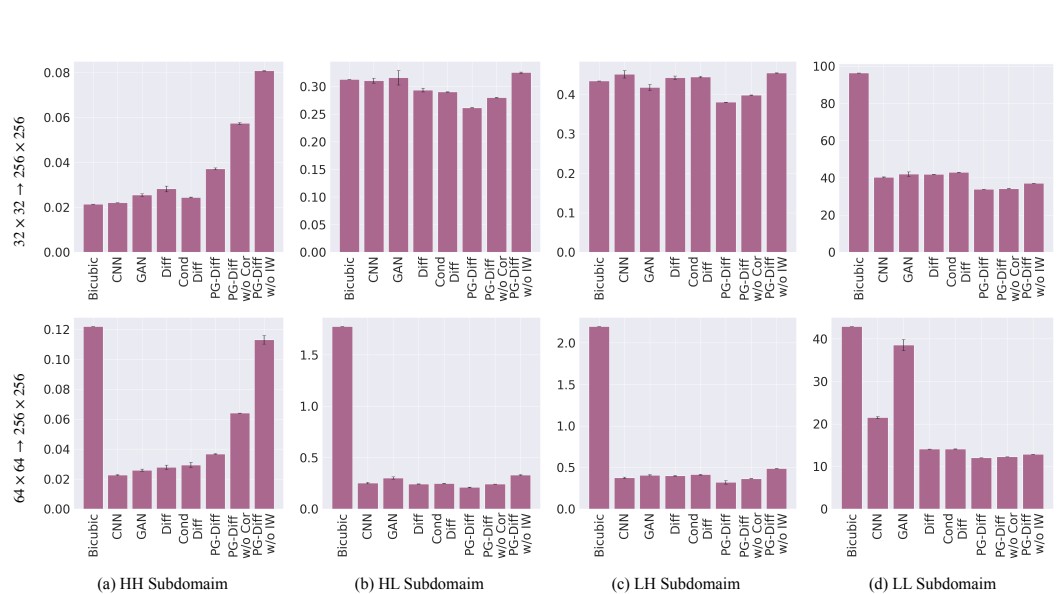

Figure 12: Wavelet subdomain L2 norm for the *Kolmogorov Flow* dataset.

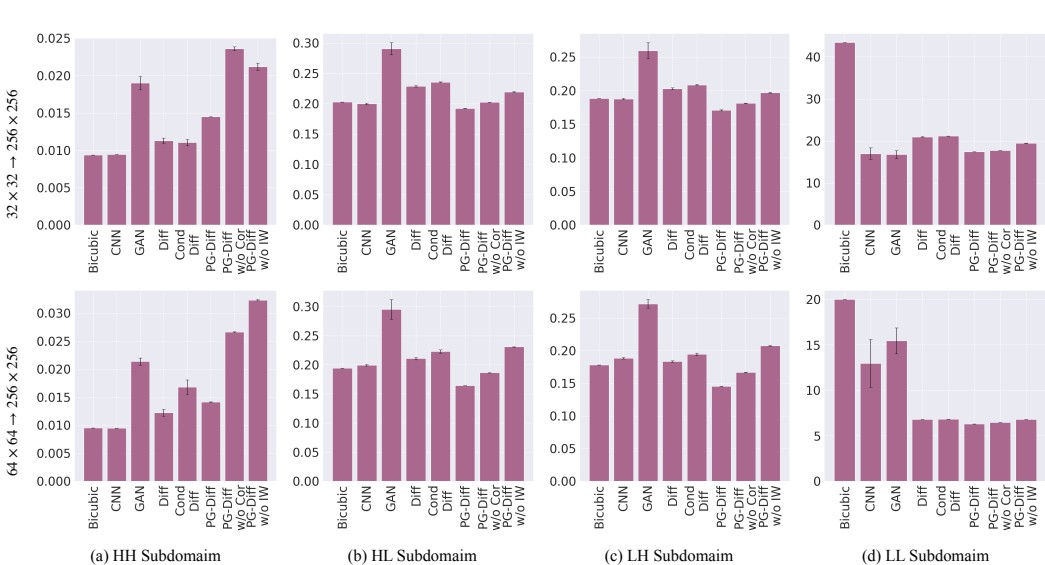

Figure 13: Wavelet subdomain L2 for the *McWilliams Flow* dataset.

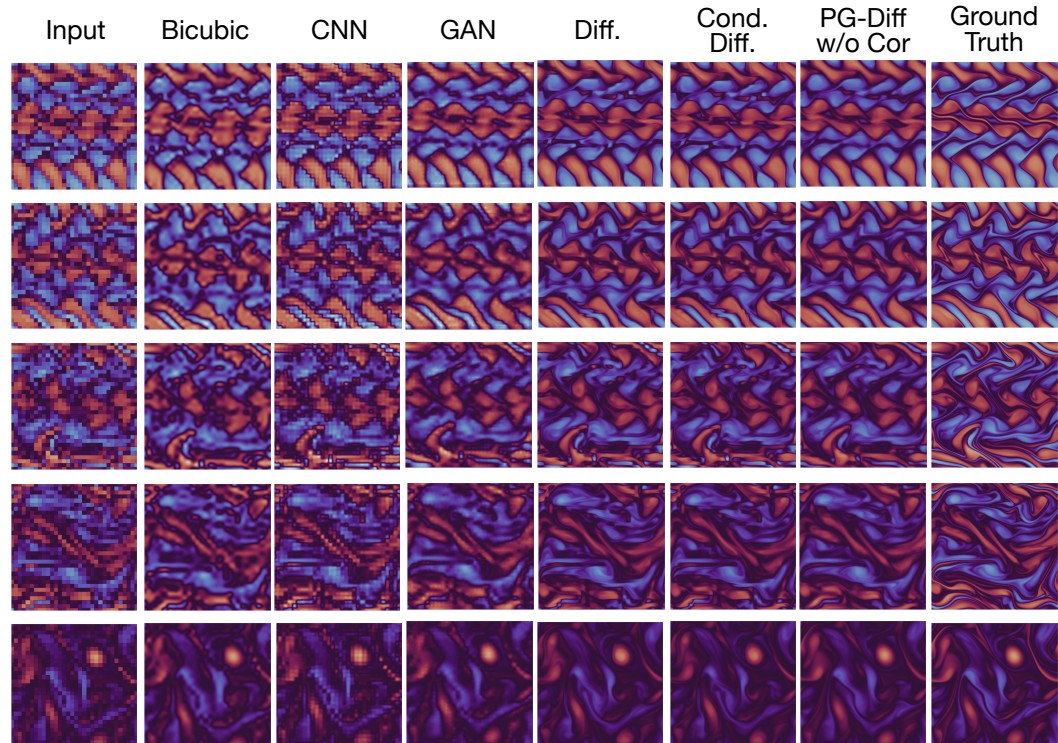

| Input | Bicubic | CNN | GAN | Diff. | Cond. Diff. | PG-Diff w/o Cor | Ground Truth |

Figure 14: Visualization comparison of reconstruction on the *Kolmogorov Flow* from PG-Diff w/o Cor. The first column displays the low-fidelity input data upscaled to a resolution of $256 \times 256$, while the last column shows the high-fidelity ground truth.

### E.4 LPIPS SCORES

Direct mapping models are effective at minimizing L2 loss. However, these models tend to produce samples that appear smooth and blurry, often missing the intricate details. This is because the L2 loss emphasizes pixel-wise accuracy, which can lead to averaging effects and the loss of fine-grained details. To address this issue, we use the LPIPS (Learned Perceptual Image Patch Similarity) metric, which focuses on perceptual differences and provides a more qualitative assessment of the reconstructed flow fields, capturing the texture and small-scale structures. The results are presented in Table 4

Although LPIPS is trained on the ImageNet dataset, which consists of natural images, it remains a valuable metric for evaluating perceptual quality in CFD applications. This is because LPIPS leverages features from deep neural networks that are effective at capturing multi-scale patterns, textures, and perceptual similarities, regardless of the specific domain. Fluid dynamics data often have complex structures and turbulent patterns that share characteristics with textures found in natural images, making LPIPS suitable for assessing the fidelity of reconstructed flow fields. Thus, despite being trained on ImageNet, LPIPS can still effectively quantify how well the reconstructed samples retain important perceptual details, making it a robust metric for evaluating the visual quality of CFD reconstructions.

### E.5 IMPORTANCE WEIGHT SENSITIVITY ANALYSIS

We present a sensitivity analysis to explore how different hyperparameter settings affect the performance of the Importance Weight Strategy. The importance weight strategy is governed by three main parameters: $\beta$ (the maximum importance weight), $\alpha$ (the minimum importance weight), and $\theta$ (the importance threshold). We adjust these parameters sequentially. The results are summarized in Figure 16.

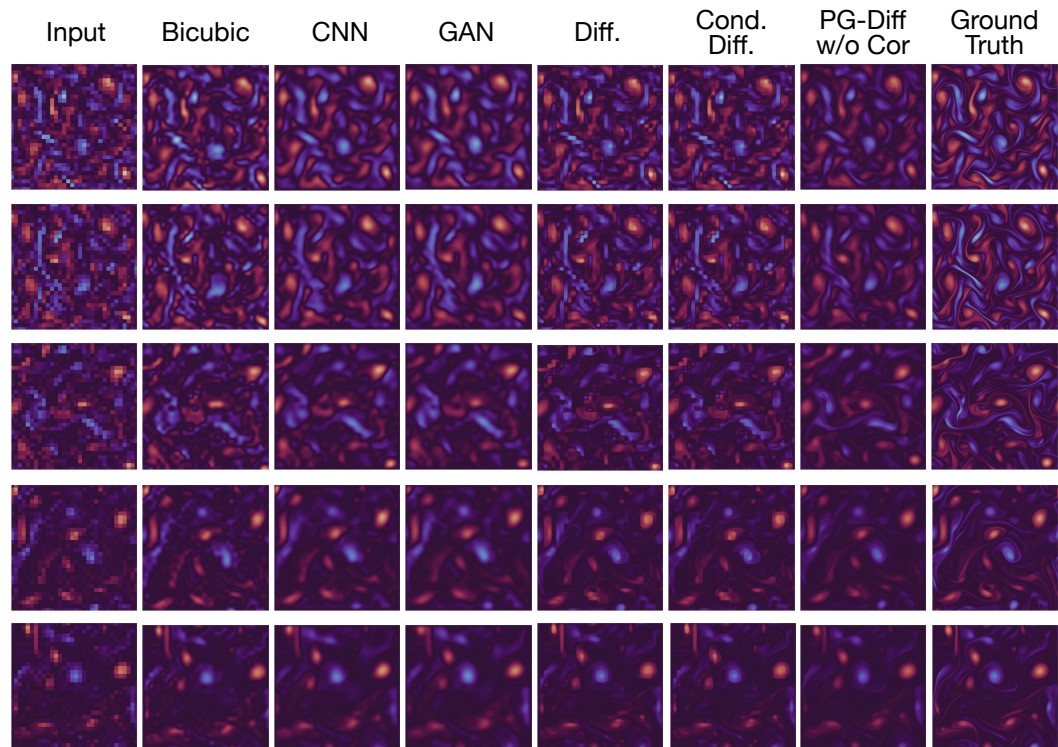

|  | Input | Bicubic | CNN | GAN | Diff. | Cond. Diff. | PG-Diff w/o Cor | Ground Truth |
|--|--|--|--|--|--|--|--|--|

Figure 15: Visualization comparison of reconstruction on the *McWilliams Flow* from PG-Diff w/o Cor. The first column displays the low-fidelity input data upscaled to a resolution of $256 \times 256$, while the last column shows the high-fidelity ground truth.

|  | Bicubic | CNN | GAN | Diff | Cond Diff | PG-Diff |
|--|--|--|--|--|--|--|
| *Kolmogorov Flow* | 0.5421 | 0.5358 | 0.5109 | 0.2848 | 0.2869 | 0.2781 |
|  | 0.2919 | 0.4065 | 0.4082 | 0.1215 | 0.1229 | 0.1097 |
| *McWilliams Flow* | 0.5736 | 0.4921 | 0.4505 | 0.3524 | 0.3540 | 0.2936 |
|  | 0.3417 | 0.3948 | 0.3948 | 0.1457 | 0.1604 | 0.1298 |
| *Decaying Turbulence* | 0.2221 | 0.1448 | 0.1991 | 0.4215 | 0.1688 | 0.1397 |
|  | 0.0794 | 0.1521 | 0.1407 | 0.3608 | 0.2259 | 0.0637 |
| *Taylor Green Vortex* | 0.3715 | 0.3331 | 0.4175 | 0.2525 | 0.1750 | 0.1704 |
|  | 0.1594 | 0.3057 | 0.3137 | 0.1494 | 0.2362 | 0.1339 |

Table 4: LPIPS scores for each dataset. Metrics are reported for both 32×32 → 256×256 ( grey ) and 64×64 → 256×256 tasks.

### E.6 COMPARISON OF RESIDUAL CORRECTION AND POST PROCESSING

We compared the performance of residual correction and post-processing technique in Table 5. The post-processing applies a single residual correction to the reconstructed high-fidelity samples in the end. The results indicate that while post-processing achieves very low PDE residuals, it significantly increases the L2 loss. In contrast, our Residual Correction method strikes a better balance between L2 loss and PDE residual.

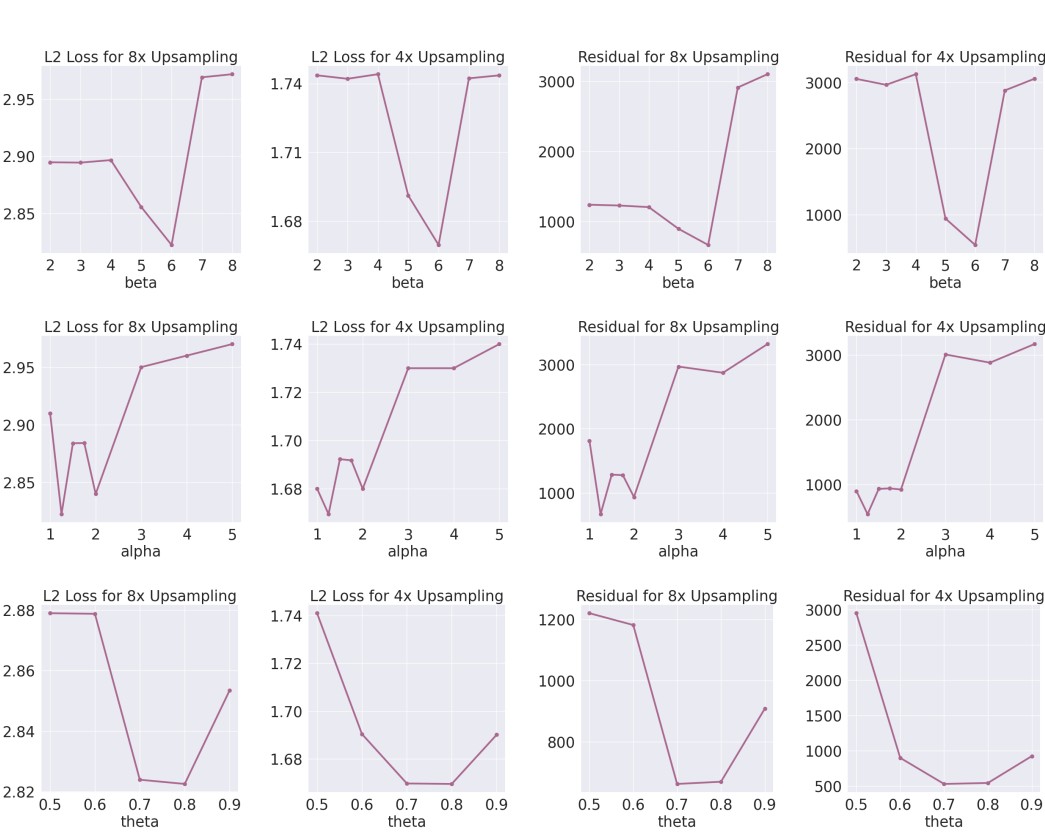

Figure 16: Sensitivity analysis of key parameters for the *Importance Weight*. Experiments are conducted on the *Kolmogorov Flow*. The top row presents the results for the maximum importance weight, $\beta$. The middle row displays the results for the minimum importance weight, $\alpha$, and the bottom row shows the results for the importance threshold, $\theta$. These three hyperparameters were tuned in sequence, and the optimal combination ($\beta = 6$, $\alpha = 1.25$, and $\theta = 0.8$) is selected.

| | 32×32 → 256×256 | | 64×64 → 256×256 | |
|---|---|---|---|---|
| | L2 | PDE Residual | L2 | PDE Residual |
| Residual Correction | 2.7897 | 40.12 | 1.6609 | 39.97 |
| Post Processing | 3.0019 | 5.42 | 1.7737 | 7.16 |

Table 5: Comparison between our proposed Residual Correction and Post Processing methods. Experiments are conducted on the *Kolmogorov Flow*.

### E.7 GENERALIZATION

We show the generalization ability of our model across various settings. The original *Kolmogorov Flow* dataset is generated using timestep $dt = 1/32$, spatial domain size $2\pi \times 2\pi$, and Reynolds number $Re = 1000$.

Table 6: Generalization results on *Kolmogorov Flow* dataset with $64 \times 64 \to 256 \times 256$ setting.

| Variation | Model | L2 | PDE Residual |
|---|---|---|---|
| **Time Discretization Variations** | | | |
| $dt = 1/40$ | Trained on Original Data | 1.6782 | 31.49 |
| | Trained on $dt = 1/40$ Data | 1.6723 | 30.91 |
| $dt = 1/50$ | Trained on Original Data | 1.6484 | 33.49 |
| | Trained on $dt = 1/50$ Data | 1.6489 | 26.66 |
| **Spatial Domain Size Variations** | | | |
| Domain Size $1\pi \times 1\pi$ | Trained on Original Data | 1.0255 | 263.23 |
| | Trained on $1\pi \times 1\pi$ Data | 0.9443 | 207.03 |
| Domain Size $1.5\pi \times 1.5\pi$ | Trained on Original Data | 1.3212 | 84.24 |
| | Trained on $1.5\pi \times 1.5\pi$ Data | 1.2918 | 94.87 |
| **Reynolds Number Variations** | | | |
| $Re = 500$ | Trained on Original Data | 1.3112 | 21.57 |
| | Trained on $Re = 500$ Data | 1.2998 | 22.96 |
| $Re = 2000$ | Trained on Original Data | 1.9647 | 24.07 |
| | Trained on $Re = 2000$ Data | 1.9694 | 24.22 |

### E.8 RUNTIME COMPARISON

Table 9 presents the time required for the numerical solver to generate a single frame of low- and high-fidelity samples for each dataset. The results indicate that generating even $64 \times 64$ low-fidelity data is significantly faster than producing high-fidelity data, emphasizing the value of using ML approach for accelerating high-fidelity simulations.

| Dataset | $256 \times 256$ | $64 \times 64$ | $32 \times 32$ |
|---|---|---|---|
| *Kolmogorov Flow* | 138.49 | 0.44 | 0.04 |
| *McWilliams Flow* | 98.99 | 1.23 | 0.12 |
| *Decaying Turbulence* | 93.82 | 1.01 | 0.06 |
| *Taylor Green Vortex* | 139.35 | 1.49 | 0.22 |

Table 7: Run time of numerical solver to generate one frame for each dataset across different grid resolutions with a batch size of 10. All times are measured in seconds.

Table 8 shows the time comparison across ML models for reconstructing high-fidelity data from low-fidelity inputs.

| Method | $32{\times}32 \rightarrow 256{\times}256$ | $64{\times}64 \rightarrow 256{\times}256$ |
|---|---|---|
| Bicubic | 1.43e-5 | 1.4e-5 |
| CNN | 0.005 | 0.005 |
| GAN | 0.008 | 0.008 |
| Diff | 6.10 | 3.06 |
| Cond Diff | 6.20 | 3.38 |
| PG-Diff | 6.37 | 3.27 |
| PG-Diff w/o Cor | 6.10 | 3.06 |
| PG-Diff w/o IW | 6.37 | 3.27 |

Table 8: Runtime comparison of various methods across different resolution levels using a batch size of 10, with all times measured in seconds. These results are based on the *McWilliams Flow* dataset.

| Dataset | Same error with $64 \times 64$ | Same error with $32 \times 32$ |
|---|---|---|
| *Kolmogorov Flow* | 35.36 | 15.15 |
| *McWilliams Flow* | 39.08 | 17.81 |
| *Decaying Turbulence* | 21.51 | 17.41 |
| *Taylor Green Vortex* | 33.23 | 15.49 |

Table 9: Run time of numerical solver to generate one frame for each dataset with approximately same error compared to PG-Diff with a batch size of 10. All times are measured in seconds.

## E.9 EMBEDDING SUPER RESOLUTION WITHIN SOLVER

For longer rollouts on coarser grids, low-fidelity data becomes qualitatively different from high-fidelity data, making simple postprocessing with super-resolution models insufficient to recover the trajectory. However, we demonstrate that when integrated into the solver, PG-Diffenhances numerical simulations on coarse grids. Our approach follows the pipeline: "solver $\rightarrow$ super resolution $\rightarrow$ downsample $\rightarrow$ solver." Specifically, we begin with a numerical solver to perform one-step predictions on a coarse grid. Next, we apply PG-Diff for super-resolution, downsample the output back to the coarse grid, and use it as input for the next simulation step. We present the visual results in Figure 17.

## E.10 PSNR AND SSIM RESULTS

Table 10: Kolmogorov Flow

| Model | 4x Upsampling | | 8x Upsampling | |
|---|---|---|---|---|
| | PSNR | SSIM | PSNR | SSIM |
| Bicubic | 21.1257 | 0.4769 | 18.3063 | 0.2479 |
| CNN | 24.8310 | 0.5190 | 22.9806 | 0.3712 |
| GAN | 20.6210 | 0.4160 | 20.2323 | 0.3695 |
| Diffusion | 25.4049 | 0.6487 | 21.5818 | 0.4072 |
| Conditional Diffusion | 25.2389 | 0.6456 | 20.2067 | 0.3425 |
| PG-Diff | **26.1733** | **0.6781** | **24.0754** | **0.4409** |

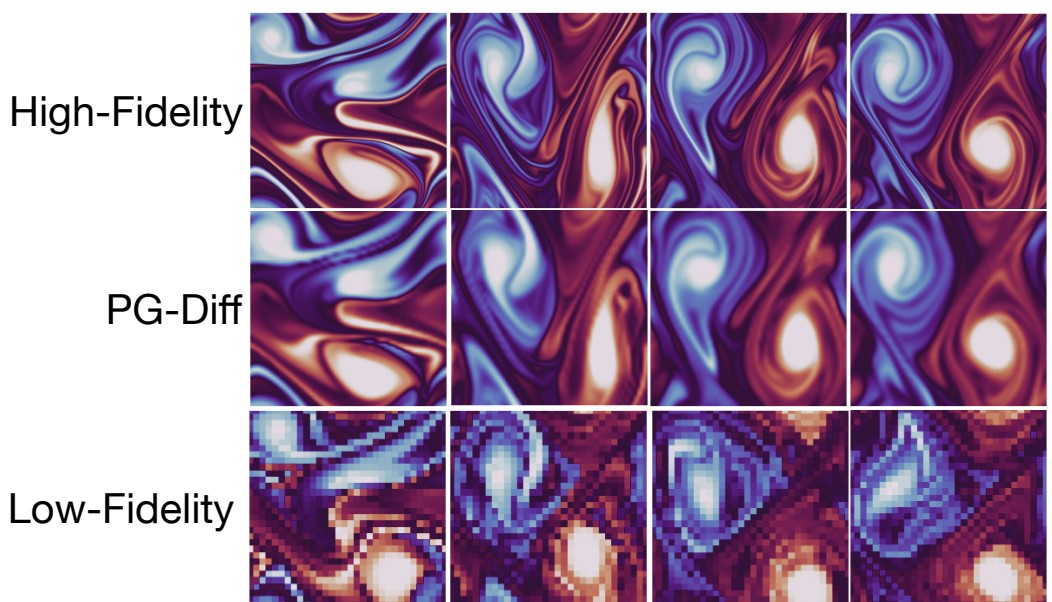

Figure 17: Visualization of reconstructed high-fidelity data when PG-Diffis integrated within the numerical solver.

Table 11: McWilliams Flow

| Model | 4x Upsampling | | 8x Upsampling | |
|---|---|---|---|---|
| | PSNR | SSIM | PSNR | SSIM |
| Bicubic | 25.1992 | 0.4834 | 21.9313 | 0.2519 |
| CNN | 28.6248 | 0.5897 | 27.4487 | **0.5164** |
| GAN | 28.6720 | 0.4621 | 27.7062 | 0.4477 |
| Diffusion | 29.71018 | 0.6686 | 25.2200 | 0.3884 |
| Conditional Diffusion | 29.6757 | 0.6665 | 25.1475 | 0.3823 |
| PG-Diff | **30.0540** | **0.6722** | **28.1972** | 0.4643 |

Table 12: Taylor Green Vortex

| Model | 4x Upsampling | | 8x Upsampling | |
|---|---|---|---|---|
| | PSNR | SSIM | PSNR | SSIM |
| Bicubic | 25.4811 | 0.7211 | 19.4033 | 0.5041 |
| CNN | 26.6066 | 0.7657 | 26.3448 | 0.7422 |
| GAN | 28.1370 | 0.7597 | 25.0551 | **0.7633** |
| Diffusion | 30.7706 | 0.8438 | 24.8698 | 0.6678 |
| Conditional Diffusion | 23.2154 | 0.6767 | 21.3979 | 0.5959 |
| PG-Diff | **31.5277** | **0.8713** | **26.7503** | 0.7562 |

Table 13: Decaying Turbulence

| Model | 4x Upsampling | | 8x Upsampling | |
|---|---|---|---|---|
| | PSNR | SSIM | PSNR | SSIM |
| Bicubic | 41.7820 | 0.8888 | 35.1032 | 0.7098 |
| CNN | 41.0336 | 0.8849 | 42.1580 | 0.9200 |
| GAN | 39.2718 | 0.9055 | 38.9931 | 0.8863 |
| Diffusion | 47.2289 | 0.9571 | 40.8901 | 0.8622 |
| Conditional Diffusion | 46.0675 | 0.9138 | 40.1089 | 0.8456 |
| PG-Diff | **48.0321** | **0.9601** | **43.0276** | **0.9295** |

