# OpenReview forum: "Physics-Informed Self-Guided Diffusion Model for High-Fidelity Simulations"
_ICLR.cc/2025/Conference — Submitted to ICLR 2025_

### Official Review · Reviewer_qB2f · 2024-10-31

**Soundness:** 3
**Presentation:** 3
**Contribution:** 2
**Rating:** 5
**Confidence:** 4

**Summary:**

This paper proposes a diffusion model designed to reconstruct high-fidelity computational fluid dynamics (CFD) data from low-fidelity solver-generated inputs. Traditional machine learning models for CFD rely on low-fidelity data artificially downsampled from high-fidelity sources, which limits performance in real-world applications. This paper trains on the data run at two different grid sizes and proposes two directions to improve upon this:

- Importance Weight Strategy during training: It uses wavelet transformation to assign importance to high-frequency flow field components, guiding the model toward better reconstruction of detailed structures.
- Residual Correction during inference: This physics-informed module applies corrections based on governing equations (e.g., Navier-Stokes) to ensure physical accuracy, especially for turbulent flows.

The model is evaluated on four datasets, generated by the incompressible Navier-Stokes equations.

**Strengths:**

This paper points out an important issue with training ML models for super-resolving computational fluid dynamics (CFD) which is when running CFD at two different grid sizes, the dynamics between them can diverge and will not correspond to an interpolation of each other.  This paper argues that one should train ML models from simulations run at different resolutions.

The paper is very nicely written. There is a sufficient number of figures to explain the core ideas.

I found the idea of using residual error correction an interesting development that makes sure that the generated high-fidelity solution matches with the underlying PDE.

**Weaknesses:**

The weighting function in 6 seems arbitrary and hacky. There is no intuitive explanation of why this form of weighting mechanism is needed. Note that this mechanism should be discussed in the context of prior work such as simple diffusion [1] who also suggest reweighting the objective based on the resolution of the data.

The paper discusses the fact that running PDEs at different resolutions will not correspond to the same solutions that are interpolated. However, it is known that running PDEs at different grid sizes will eventually diverge over time to the point that we cannot recover the high-fidelity solution from the low-fidelity input. This work does not discuss how one can correct the low-fidelity simulation with the information received from the high-fidelity solution.


[1] Simple diffusion: End-to-end diffusion for high resolution images, by Hoogeboom.

**Questions:**

In Section 3, it is stated that the reverse process is started from $x_t{_{guide}}$. How do you choose t here? How do you ensure that adding noise does not wash out the necessary signal for high-fidelity output generation?

R introduced in Algorithm 1 is central to the residual correction idea but it has not been defined properly.

I am a bit surprised to see that the diffusion model outperforms conditional diffusion models in table 1. Several works including [2] have used conditional diffusion models for this? How is the diffusion baseline set up?

Table 1 is very hard to read with all scientific number notation (with e). I would recommend using fixed decimal points.



[2] Residual Corrective Diffusion Modeling for Km-scale Atmospheric Downscaling, Mardani et al. 2023

---

> ### Comment · Area_Chair_8FEW · 2024-11-15
> **Comparison with statistical downscaling references**
>
> Dear Reviewer qB2f,
>
> As you have the most enthusiastic review, and given that you suggested the statistical downscaling baseline, how would you compare this work with https://openreview.net/forum?id=5NxJuc0T1P, as both seem to tackle a similar problem (although the  language seems to be quite different).
>
> In addition, this work https://journals.ametsoc.org/view/journals/aies/3/2/AIES-D-23-0039.1.xml, also seems relevant for the current paper.
>
> Thank you again for your time and expertise!
>
> Best,
>
> AC.

---

> ### Author Response · Authors · 2024-11-20
> **Response to Reviewer qB2f**
>
> We would like to thank the reviewer for the feedback. In what follows, we hope to address any concerns you might have.
>
> ### **[Importance Weight Weighting Function]**
> Equation 6 linearly maps the high frequency signals to an importance weight between alpha and beta. The weight in [2] is adjusted as a result of adjusting noise schedules for higher resolution images. Our attention weight is sample dependent to ensure the model captures high-fidelity fine-grained details.
>
> ### **[t_guide in Reverse Process]**
> We adopt the same t_guide for 4x upsampling and 8x upsampling as [1]. We clarified hyperparameters in Appendix C2 IMPLEMENTATION DETAILS. [1] conducted extensive studies to determine the optimal t_guide to ensure the correct amount of noise is injected to low-fidelity data. We also conducted informal experiments to verify that the t_guide suggested in [1] still works very well in our case.
>
> ### **[Residual Calculation]**
> The calculation of residual is defined in Appendix D.1 RESIDUAL CALCULATION. We also referenced this section in the main paper.
>
> ### **[Conditional Diffusion Baseline]**
> The conditional diffusion baseline follows [1]. At each backward diffusion step, we calculate the gradient of the residual with respect to the noised sample as the condition and concatenate it with the noised sample into our denoiser, a Unet, to predict noise. Our residual correction, on the other hand, utilizes the gradient of the residual with respect to **predicted cleaned samples** during backward diffusion. An intuitive interpretation for superior performance of our method is that the baseline conditional diffusion model projects the noised sample onto the solution subspace of the PDE during denoising, while our method directly projects the predicted cleaned sample onto the solution subspace of the governing PDE at **selected backward diffusion steps**. Placing two residual corrections at the beginning quickly projects the sample onto the solution subspace of the PDE to guide the backward diffusion process and two residual corrections at the end guarantees physical consistency of the reconstructed samples.
>
> ### **[Qualitatively Different Corrections]**
> Since PG-Diff only requires high-fidelity data during training and does not learn specific low-fidelity and high-fidelity relationships, it is unable to correct frames that are qualitatively incorrect as a postprocessing method. We ensured that the low-fidelity data in current experiments are qualitatively correct. However, we show that when integrated within the solver, PG-Diff can address this challenge. We present the results in the Appendix of our modified paper.
>
> The method follows “solver->super resolution->downsample->solver” We first use a numerical solver for one-step predictions at a coarse grid. Then, we apply PG-Diff for super resolution, and then downsample it to a coarse grid, which is used as input for next step simulation.
>
> We would like to highlight that even though at some point in the numerical simulation, PG-Diff will not be able to recover the trajectory, a core benefit of PG-Diff and [1] is that it only requires high-fidelity data during training. When presented with a new type of low-fidelity data, for example 40x40, PG-Diff only needs a small amount of validation dataset to determine the optimal t_guide. Then the pre-trained model can be directly applied to reconstruct from the low-fidelity data.
>
> [1] A physics-informed diffusion model for high-fidelity flow field reconstruction.\
> [2] Machine learning accelerated computational fluid dynamics.

---

> > ### Comment · Reviewer_qB2f · 2024-12-02
> > **Thanks for the rebuttal**
> >
> > As other reviewers pointed out, the problem of diverging PDE solutions at different scales has already been discussed in previous papers. The idea of reweighting different resolutions and using wavelet-based decomposition was explored in prior works such as simple diffusion. Given these, this paper does not propose a significantly new perspective on the problem, and as a result, I've decided to reduce my rating to 5 from 6.

---

> ### Author Response · Authors · 2024-11-25
> **Follow-up the discussion**
>
> Dear Reviewer qB2f,
>
> We greatly appreciate your time and feedback on our work. We have carefully addressed your comments and clarified potential misunderstandings. Additionally, we also included new experimental results to corroborate our findings.
>
> We kindly invite you to revisit our paper in light of these updates and clarifications. We would greatly appreciate it if you could consider whether our responses warrant a reevaluation of your rating.
>
> Best regards,
>
> The Authors

---

### Official Review · Reviewer_ZSxU · 2024-11-01

**Soundness:** 3
**Presentation:** 2
**Contribution:** 2
**Rating:** 6
**Confidence:** 4

**Summary:**

The paper proposes a diffusion-based method for reconstructing high-fidelity physics simulation data given low-fidelity input. The paper also proposes to incorporate prior knowledge via the gradient of PDE residual and a new weighting scheme based on multi-resolution analysis (Wavelet transform) for diffusion loss. The numerical experiments on several 2D flow problem showcase that the proposed method has better L2 accuracy and lower residuals over other baseline methods.

**Strengths:**

1. Upsampling and reconstructing under-resolved physics is important for building hybrid solver and inverse problem in PDE applications.

2. A new spatial weighting scheme based on Wavelet transformation which modulate the loss based on the spectrum of signal. The new scheme is technically sound and experiments show that it consistently improves model performance.

3. The introduction to the proposed method is clear and easy-to-follow.

**Weaknesses:**

1. Applying diffusion model with guidance based on target constraint for the inverse problem is not an entirely new technique (for example, DPS: https://arxiv.org/pdf/2209.14687 and On conditional diffusion models for PDE simulations: https://arxiv.org/abs/2410.16415 has explored similar technique)

2. The evaluation of model’s prediction in terms of physics coherence is relatively vague. First, all the reported residuals are quite large and there is no reference showing what is a reasonable scale. While the average error of different frequency components in the wavelet domain is shown, there is no information on the spectrum of different wavelength.

3. The authors run the solver on a coarse grid to get the “low fidelity” data and then run the solver on fine grid to get “high fidelity” data, instead of artificially downsampling the data. The low-fidelity simulation will deviate from the high-fidelity one as time evolves due to the under-resolved error. Yet looking at the figure comparing data trajectory of different fidelities, the general structures of the vortices are very similar across different fidelities (for example, Figure 8) in my opinion. I hope the authors can provide more clarification and analysis regarding the dataset, such as the spectrum of different simulation and perhaps a simple study of mesh convergence.

**Questions:**

* Following point 2, what is the residual of the reference DNS simulation?

* The authors state that the high fidelity data for 2D Kolmogorov flow is derived by running DNS on a 256 x 256 grid (feel free to correct me if I’m wrong). However, under a similar setting (Re=1000), prior works like Shu et al. [1] and Kochkov et al. [2] have used a much finer discretization, i.e. 2048 x 2048.

* (Minor) In algorithm 1, what is the rationale for using Adam instead of simpler gradient descent? In addition, is Adam re-initialized after very DDIM step?

[1] Shu, D., Li, Z., & Farimani, A. B. (2023). A physics-informed diffusion model for high-fidelity flow field reconstruction. Journal of Computational Physics, 478, 111972.

[2] Kochkov, Dmitrii, et al. "Machine learning–accelerated computational fluid dynamics." Proceedings of the National Academy of Sciences 118.21 (2021): e2101784118.

---

> ### Author Response · Authors · 2024-11-20
> **Response to Reviewer ZSxU**
>
> We would like to thank the reviewer for the feedback. In what follows, we hope to address any concerns you might have.
>
> ### **[Diffusion Model Guided Generation Literature]**
> Thank you for pointing out other related work in diffusion model guided generation. We would like to clarify our differences with these works.
>
> -[3, 4] apply guidance to the noised samples, and [6] incorporate physical guidance in the score function. Our guidance is applied to the **predicted cleaned sample**. After correcting the predicted cleaned sample, we use it to calculate the noised sample at the next timestep. [1] has proposed a similar approach to apply a single step of gradient descent to the noised sample at each backward step as a baseline model, but the performance is inferior to the conditional diffusion model, which implies that it is also inferior to PG-Diff.
>
> -The proposed solution in [5] to solve the data subproblem either uses an analytical solution or a single step of gradient descent (Equation 13). Since analytical solutions in our problem are unavailable, we leverage **multiple steps** of gradient descent with **momentum and adaptive scaling** at **selected backward diffusion steps**. We notice that a single step of gradient descent is insufficient to minimize the residual. Thus, we use Adam to consider momentum and adaptive scaling to perform multiple gradient descent steps. In addition, differing from [3,4,5], our guidance is applied at selected backward diffusion steps. Our experiments in 4.5 PHYSICAL GUIDANCE reveal that even though an excessive amount of corrections can minimize the residual, it compromises L2 loss. Thus, we apply two corrections at the start and two corrections at the end to achieve optimal balance between physical consistency(Residual) and predictive accuracy(L2 Loss).
>
> We have revised our manuscript to cite and discuss these related work.
>
> ### **[Residual of Reference DNS]**
> The residual of the reference DNS simulation is reported in the following table. For each frame, we calculate the residual to obtain a 256x256 matrix. Then, we calculate the average sum of squares as our evaluation metrics. While the residual reported in our Table 1 is larger than that in [1], [1] normalizes the average sum of squares of the residual using high-fidelity residual.
>
> We believe that normalization with high-fidelity residuals could cause controversy. While [1] argues that the residual metrics is the smaller the better, one can argue that since the high-fidelity data is the ground truth, reconstructed samples with residual closer to that of the high-fidelity pair should be better. Since existing works on CFD super resolution claim that residual metrics is the smaller the better, we adopt unnormalized residuals in our paper.
>
> We would like to point out that under the evaluation framework where residuals closer to high-fidelity pairs indicate better physical consistency, PG-Diff surpasses all baseline models in every task, since its reconstructed samples have residuals closest to that of high-fidelity pairs.
>
> |                | **Kolmogorov** | **McWilliams** | **TGV**   | **Decay Turbulence** |
> |----------------|----------------|----------------|-----------|-----------------------|
> | **Residual**   | 44.13          | 6.64           | 5101.65   | 212.43               |
>
> ### **[High-Fidelity Data Grid]**
> We apologize for the oversight when drafting the paper. The high-fidelity data generation follows [1] and uses the pseudo-spectral solver from [2]. The direct numerical simulation was performed on a 2048x2048 discretization grid and then we uniformly downsample the data to 256x256 as our high-fidelity data. We have updated our manuscript to correctly describe the high-fidelity data generation.
>
> ### **[Adam in Algorithm 1]**
> The Adam gradient descent in Algorithm 1 involves multiple steps and uses momentum and adaptive scaling. Empirically, we found that Adam performed better than vanilla gradient descent. Adam is re-initialized after every DDIM step.
>
> [1] A physics-informed diffusion model for high-fidelity flow field reconstruction.\
> [2] Physics-informed neural operator for learning partial differential equations.\
> [3] Diffusion Posterior Sampling For General Noisy Inverse Problems.\
> [4] DiffusionPDE: Generative PDE-Solving Under Partial Observation.\
> [5] Denoising Diffusion Models for Plug-and-Play Image Restoration.\
> [6] On conditional diffusion models for PDE simulations.

---

> > ### Author Response · Authors · 2024-11-20
> > **Response to Reviewer ZSxU Continued**
> >
> > ### **[Qualitatively Different Corrections]**
> > Since PG-Diff only requires high-fidelity data during training and does not learn specific low-fidelity and high-fidelity relationships, it is unable to correct frames that are qualitatively incorrect as a postprocessing method. We ensured that the low-fidelity data in current experiments are qualitatively correct. However, we show that when integrated within the solver, PG-Diff can address this challenge. We present the results in the Appendix of our modified paper.
> >
> > The method follows “solver->super resolution->downsample->solver” We first use a numerical solver for one-step predictions at a coarse grid. Then, we apply PG-Diff for super resolution, and then downsample it to a coarse grid, which is used as input for next step simulation.
> >
> > We would like to highlight that even though at some point in the numerical simulation, PG-Diff will not be able to recover the trajectory, a core benefit of PG-Diff and [1] is that it only requires high-fidelity data during training. When presented with a new type of low-fidelity data, for example 40x40, PG-Diff only needs a small amount of validation dataset to determine the optimal t_guide. Then the pre-trained model can be directly applied to reconstruct from the low-fidelity data.
> >
> > [1] A physics-informed diffusion model for high-fidelity flow field reconstruction.

---

> ### Author Response · Authors · 2024-11-25
> **Follow-up the discussion**
>
> Dear Reviewer ZSxU,
>
> We greatly appreciate your time and feedback on our work. We have carefully addressed your comments and clarified potential misunderstandings. Additionally, we also included new experimental results to corroborate our findings.
>
> We kindly invite you to revisit our paper in light of these updates and clarifications. We would greatly appreciate it if you could consider whether our responses warrant a reevaluation of your rating.
>
> Best regards,
>
> The Authors

---

> > ### Comment · Reviewer_ZSxU · 2024-11-26
> > **Reply to authors**
> >
> > I would like to thank the authors for their efforts and clarification. Some of my concerns such as the quality of the data have been addressed. I adjusted my score from 5 to 6.
> >
> > However, I think some weaknesses remain. First, conditional diffusion + learning is a quite common technique for diffusion inverse problem application. Applying it to PDE problems with residual constraint is definitely an interesting direction though. Second, putting the solver in the loop is a good start, however there is not too much quantitative evidence in the paper showing that the model do correct the deviation of the trajectory caused by under-resolved mesh unlike previous works [1, 2]. Currently, the results presented qualitatively shows that diffusion model can make the results smoother and seemingly more coherent.
> >
> > [1] Um, Kiwon, et al. "Solver-in-the-loop: Learning from differentiable physics to interact with iterative pde-solvers." Advances in Neural Information Processing Systems 33 (2020): 6111-6122.
> >
> > [2] Kochkov, Dmitrii, et al. "Machine learning–accelerated computational fluid dynamics." Proceedings of the National Academy of Sciences 118.21 (2021): e2101784118.

---

> > > ### Author Response · Authors · 2024-11-27
> > >
> > > Thank you for taking the time to review our rebuttal and for raising the score. We greatly appreciate your thoughtful feedback and support.

---

### Official Review · Reviewer_ovnW · 2024-11-03

**Soundness:** 2
**Presentation:** 3
**Contribution:** 2
**Rating:** 5
**Confidence:** 3

**Summary:**

This manuscript aims to solve the problem of super-resolution of physical fields. Specifically, they draw on the framework in [1]. The difference lies in 1) weighting the denoising loss over pixels using the wavelet transform and 2) performing physical loss gradient descent over clean predictions.

[1] A physics-informed diffusion model for high-fidelity flow field reconstruction

**Strengths:**

1. Authors present four turbulence datasets with different characteristics.

2. I agree with one of the author's points in the introductory section. They note that some of the current work focuses on low-resolution data that is downsampled. This is not consistent with reality, and in fact, to apply these super-resolution models, such data should come from the low-resolution simulation of a PDE solver, which typically has lower fidelity than downsampled ones.

2. The amount of experiments is rich enough, and all the graphs are clear.

**Weaknesses:**

1. I have some questions about the authors' contributions to modeling. First is the training phase. The authors use the wavelet transform to determine which pixels need to be weighted. As can be seen from the visualization in Figure 10, the weighting seems to be added only at high frequencies. This raises the question of whether a complex wavelet transform needs to be introduced. This is because we can simply call the Laplace operator to achieve this effect. Furthermore, from the quantitative results in Table 1, it seems that this measure does not contribute much to the results.

2. In addition, another contribution of the authors is to perform gradient descent on clean samples at each step to reduce the residuals on the equations. The authors ignore many related diffusion model guided generation literature [1,2,3] that the authors do not cite and discuss here.

3. The authors claim that their high-fidelity data is actually simulated on a grid of 256. It is important to realize that for such a large Reynolds number (1000+), this resolution is usually insufficient. shu et al. [4]'s data was simulated on a grid of 2048.

[1] DIFFUSION POSTERIOR SAMPLING FOR GENERAL NOISY INVERSE PROBLEMS

[2] Denoising Diffusion Models for Plug-and-Play Image Restoration

[3] DiffusionPDE: Generative PDE-Solving Under Partial Observation

[4] A physics-informed diffusion model for high-fidelity flow field reconstruction

**Questions:**

see Weaknesses

---

> ### Author Response · Authors · 2024-11-20
> **Response to Reviewer ovnW**
>
> We would like to thank the reviewer for the feedback. In what follows, we hope to address any concerns you might have.
>
> ### **[Wavelet Transformation vs Laplace Operator]**
> When designing the model, we compared the performance of wavelet transformation based importance weight and gradient based importance weight. Empirically, we found that wavelet transformation produced better results. We report the performance of PG-Diff with wavelet transformation based, gradient based, or laplacian based importance weight only(Residual Correction is not included) in the following table. The experiments are conducted on Kolmogorov Flow and tested on validation dataset. We hypothesize that the better performance of wavelet transformation based importance weight is due to its better assignment of $a_{i,j}$.
>
> |                      | 4x Upsampling      |                     | 8x Upsampling      |                     |
> |----------------------|--------------------|---------------------|--------------------|---------------------|
> |                      | **L2**            | **Residual**        | **L2**            | **Residual**        |
> | **Wavelet**          | 1.7363            | 526.62              | 2.9029            | 661.45              |
> | **Gradient**         | 1.7603            | 897.38              | 3.0168            | 935.46              |
> | **Laplacian**        | 1.7510            | 703.71              | 2.9672            | 681.45              |
>
> Additionally, in Table 1, we presented PG-Diff with both Importance Weight and Residual Correction as well as two ablation studies PG-Diff w/o Cor (Diffusion model with Importance Weight only) and PG-Diff w/o IW (Diffusion model with Residual Correction only). The results from PG-Diff and PG-Diff w/o Cor consistently show that Importance Weight reduces the L2 loss by a margin and Residual Correction crucially aids in physical consistency. Thus, we believe Importance Weight is a useful module.
>
> ### **[Diffusion Model Guided Generation Literature]**
> Thank you for pointing out other related work in diffusion model guided generation. We would like to clarify our differences with these works.
>
> -[3, 4] apply guidance to the noised samples, and [6] incorporate physical guidance in the score function. Our guidance is applied to the **predicted cleaned sample**. After correcting the predicted cleaned sample, we use it to calculate the noised sample for the next timestep. [1] has proposed a similar approach to apply a single step of gradient descent to the noised sample at each backward step as a baseline model, but the performance is inferior to the conditional diffusion model, which implies that it is also inferior to PG-Diff.
>
> -The proposed solution in [5] to solve the data subproblem either uses an analytical solution or a single step of gradient descent (Equation 13). Since analytical solutions in our problem are unavailable, we leverage **multiple steps** of gradient descent with **momentum and adaptive scaling** at **selected backward diffusion steps**. We notice that a single step of gradient descent is insufficient to minimize the residual. Thus, we use Adam to consider momentum and adaptive scaling to perform multiple gradient descent steps. In addition, differing from [3,4,5], our guidance is applied at selected backward diffusion steps. Our experiments in 4.5 PHYSICAL GUIDANCE reveal that even though an excessive amount of corrections can minimize the residual, it compromises L2 loss. Thus, we apply two corrections at the start and two corrections at the end to achieve optimal balance between physical consistency(Residual) and predictive accuracy(L2 Loss).
>
> We have revised our manuscript to cite and discuss these related works.
>
> ### **[High-Fidelity Data Grid]**
> We apologize for the oversight when drafting the paper. The high-fidelity data generation follows [1] and uses the pseudo-spectral solver from [2]. The direct numerical simulation was performed on a 2048x2048 discretization grid and then we uniformly downsample the data to 256x256 as our high-fidelity data. We have updated our manuscript to correctly describe the high-fidelity data generation.
>
> [1] A physics-informed diffusion model for high-fidelity flow field reconstruction.\
> [2] Physics-informed neural operator for learning partial differential equations.\
> [3] Diffusion Posterior Sampling For General Noisy Inverse Problems.\
> [4] DiffusionPDE: Generative PDE-Solving Under Partial Observation.\
> [5] Denoising Diffusion Models for Plug-and-Play Image Restoration.\
> [6] On conditional diffusion models for PDE simulations.

---

> ### Author Response · Authors · 2024-11-25
> **Follow-up the discussion**
>
> Dear Reviewer ovnW,
>
> We greatly appreciate your time and feedback on our work. We have carefully addressed your comments and clarified potential misunderstandings. Additionally, we also included new experimental results to corroborate our findings.
>
> We kindly invite you to revisit our paper in light of these updates and clarifications. We would greatly appreciate it if you could consider whether our responses warrant a reevaluation of your rating.
>
> Best regards,
>
> The Authors

---

> > ### Comment · Reviewer_ovnW · 2024-11-26
> >
> > Thanks for addressing my concerns. I am considering other reviewers' comments carefully before the final score recommendation.

---

> > > ### Author Response · Authors · 2024-11-27
> > >
> > > Thank you for your thoughtful consideration and for reviewing our rebuttal. We appreciate your careful evaluation and look forward to your final recommendation.

---

### Official Review · Reviewer_2FBP · 2024-11-03

**Soundness:** 2
**Presentation:** 3
**Contribution:** 2
**Rating:** 6
**Confidence:** 4

**Summary:**

This paper proposes PG-Diff, a diffusion model for high-fidelity flow field reconstruction. They introduce importance weight strategy during training as self-guidance and a training-free residual connection method during inference as physical inductive bias, to overcome the distribution shift challenge in this task.

**Strengths:**

- This paper found that current SOTA reconstruction models fail to generate high-quality outputs due to large distribution shifts between low- and high-fidelity data.
- To address this issue, they propose a diffusion model to reconstruct the high-quality outputs through the guidance of an Importance Weight strategy during training as self-guidance and
a training-free Residual Correction method during inference as physical inductive bias.

**Weaknesses:**

- Literature review is not comprehensive, some papers in solving high-fidelity fluid flow reconstruction are not properly cited: \
[1] “Fu, Cong, Jacob Helwig, and Shuiwang Ji. "Semi-Supervised Learning for High-Fidelity Fluid Flow Reconstruction." Learning on Graphs Conference. PMLR, 2024.” \
[2] “Esmaeilzadeh, Soheil, et al. "Meshfreeflownet: A physics-constrained deep continuous space-time super-resolution framework." SC20: International Conference for High Performance Computing, Networking, Storage and Analysis. IEEE, 2020.” \
[3] “Ren, Pu, et al. "PhySR: Physics-informed deep super-resolution for spatiotemporal data." Journal of Computational Physics 492 (2023): 112438.” \
[4] “Gao, Han, Luning Sun, and Jian-Xun Wang. "Super-resolution and denoising of fluid flow using physics-informed convolutional neural networks without high-resolution labels." Physics of Fluids 33.7 (2021).” \

- For comprehensive evaluation, I recommend authors also add commonly used metrics from image super-resolution such as PSNR and SSIM.
- For Figure 4, the high-resolution and low-resolution simulations basically have very similar pattern, which doesn’t well demonstrate the assumption that coarse-grained simulation differ from high-resolution simulation. I recommend authors use the test equation in this paper (“Machine learning accelerated computational fluid dynamics”: https://arxiv.org/pdf/2102.01010), In Fig 2 of that paper, it’s clear at time step 1500 that fluid flow simulated from low-resolution input deviate a lot from high-resolution input, where simple super-resolution cannot be applied. I would like to see PG-Diff can reconstruct high-fidelity flow in this scenario.

**Questions:**

- In algorithm 1, during inference the input is not pure noise, then how to determine T_guide?

---

> ### Author Response · Authors · 2024-11-20
> **Response to Reviewer 2FBP**
>
> We would like to thank the reviewer for the feedback. In what follows, we hope to address any concerns you might have.
>
> ### **[Literature Review]**
> Thank you for pointing out other works on high-fidelity fluid flow reconstruction. We have updated Section 2.1 AI for Computational Fluid Dynamics (CFD) to discuss these related works.
>
> ### **[PSNR and SSIM Results]**
> We report the PSNR and SSIM results in the following tables. PG-Diff outperforms baseline models in almost every case.
>
> ### Kolmogorov
>
> | Model                 | 4x Upsampling      |                     | 8x Upsampling      |                     |
> |-----------------------|--------------------|---------------------|--------------------|---------------------|
> |                       | **PSNR**          | **SSIM**            | **PSNR**          | **SSIM**            |
> | **Bicubic**           | 21.1257           | 0.4769              | 18.3063           | 0.2479              |
> | **CNN**               | 24.8310           | 0.5190              | 22.9806           | 0.3712              |
> | **GAN**               | 20.6210           | 0.4160              | 20.2323           | 0.3695              |
> | **Diffusion**         | 25.4049           | 0.6487              | 21.5818           | 0.4072              |
> | **Conditional Diffusion** | 25.2389       | 0.6456              | 20.2067           | 0.3425              |
> | **PG-Diff**           | **26.1733**       | **0.6781**          | **24.0754**       | **0.4409**          |
>
> ### McWilliams
>
> | Model                 | 4x Upsampling      |                     | 8x Upsampling      |                     |
> |-----------------------|--------------------|---------------------|--------------------|---------------------|
> |                       | **PSNR**          | **SSIM**            | **PSNR**          | **SSIM**            |
> | **Bicubic**           | 25.1992           | 0.4834              | 21.9313           | 0.2519              |
> | **CNN**               | 28.6248           | 0.5897              | 27.4487           | **0.5164**          |
> | **GAN**               | 28.6720           | 0.4621              | 27.7062           | 0.4477              |
> | **Diffusion**         | 29.71018          | 0.6686              | 25.2200           | 0.3884              |
> | **Conditional Diffusion** | 29.6757       | 0.6665              | 25.1475           | 0.3823              |
> | **PG-Diff**           | **30.0540**       | **0.6722**          | **28.1972**       | 0.4643              |
>
> ### Taylor Green Vortex
>
> | Model                 | 4x Upsampling      |                     | 8x Upsampling      |                     |
> |-----------------------|--------------------|---------------------|--------------------|---------------------|
> |                       | **PSNR**          | **SSIM**            | **PSNR**          | **SSIM**            |
> | **Bicubic**           | 25.4811           | 0.7211              | 19.4033           | 0.5041              |
> | **CNN**               | 26.6066           | 0.7657              | 26.3448           | 0.7422              |
> | **GAN**               | 28.1370           | 0.7597              | 25.0551           | **0.7633**          |
> | **Diffusion**         | 30.7706           | 0.8438              | 24.8698           | 0.6678              |
> | **Conditional Diffusion** | 23.2154       | 0.6767              | 21.3979           | 0.5959              |
> | **PG-Diff**           | **31.5277**       | **0.8713**          | **26.7503**       | 0.7562              |
>
> ### Decay Turbulence
>
> | Model                 | 4x Upsampling      |                     | 8x Upsampling      |                     |
> |-----------------------|--------------------|---------------------|--------------------|---------------------|
> |                       | **PSNR**          | **SSIM**            | **PSNR**          | **SSIM**            |
> | **Bicubic**           | 41.7820           | 0.8888              | 35.1032           | 0.7098              |
> | **CNN**               | 41.0336           | 0.8849              | 42.1580           | 0.9200              |
> | **GAN**               | 39.2718           | 0.9055              | 38.9931           | 0.8863              |
> | **Diffusion**         | 47.2289           | 0.9571              | 40.8901           | 0.8622              |
> | **Conditional Diffusion** | 46.0675       | 0.9138              | 40.1089           | 0.8456              |
> | **PG-Diff**           | **48.0321**       | **0.9601**          | **43.0276**       | **0.9295**          |

---

> > ### Author Response · Authors · 2024-11-20
> > **Response to Reviewer 2FBP Continued**
> >
> > ### **[Qualitatively Different Corrections]**
> > Since PG-Diff only requires high-fidelity data during training and does not learn specific low-fidelity and high-fidelity relationships, it is unable to correct frames that are qualitatively incorrect as a postprocessing method. We ensured that the low-fidelity data in current experiments are qualitatively correct. However, we show that when integrated within the solver, PG-Diff can address this challenge. We present the results in the Appendix of our modified paper.
> >
> > The method follows “solver->super resolution->downsample->solver” We first use a numerical solver for one-step predictions at a coarse grid. Then, we apply PG-Diff for super resolution, and then downsample it to a coarse grid, which is used as input for next step simulation.
> >
> > We would like to highlight that even though at some point in the numerical simulation, PG-Diff will not be able to recover the trajectory, a core benefit of PG-Diff and [1] is that it only requires high-fidelity data during training. When presented with a new type of low-fidelity data, for example 40x40, PG-Diff only needs a small amount of validation dataset to determine the optimal t_guide. Then the pre-trained model can be directly applied to reconstruct from the low-fidelity data.
> >
> > ### **[t_guide in inference]**
> > We adopt the same t_guide for 4x upsampling and 8x upsampling as [1]. We clarified hyperparameters in Appendix C2 IMPLEMENTATION DETAILS. We also conducted informal experiments to verify that the t_guide suggested in [1] still works very well in our case.
> >
> > [1] A physics-informed diffusion model for high-fidelity flow field reconstruction. \
> > [2] Machine learning accelerated computational fluid dynamics.

---

> ### Author Response · Authors · 2024-11-25
> **Follow-up the discussion**
>
> Dear Reviewer 2FBP,
>
> We greatly appreciate your time and feedback on our work. We have carefully addressed your comments and clarified potential misunderstandings. Additionally, we also included new experimental results to corroborate our findings.
>
> We kindly invite you to revisit our paper in light of these updates and clarifications. We would greatly appreciate it if you could consider whether our responses warrant a reevaluation of your rating.
>
> Best regards,
>
> The Authors

---

> > ### Comment · Reviewer_2FBP · 2024-11-27
> >
> > Thanks author for the effort in rebuttal. My concerns are well addressed, so I've increased my score from 5 to 6.

---

> > > ### Author Response · Authors · 2024-11-27
> > >
> > > Thank you for taking the time to review our rebuttal and for raising the score. We greatly appreciate your thoughtful feedback and support.

---

### Official Review · Reviewer_Vjib · 2024-11-04

**Soundness:** 2
**Presentation:** 1
**Contribution:** 2
**Rating:** 3
**Confidence:** 3

**Summary:**

This paper proposes PG-Diff which uses diffusion model to generate high-fidelity computational fluid dynamics (CFD) data when given low-fidelity CFD data. To make it works, during the training phase, the authors introduce "Importance Weight" to the loss function; during the inference phase, the authors introduce "Residual Correction" as physical guidance.

**Strengths:**

This paper leverages diffusion model to solve problems in fluid dynamics. One advantage of this current version is that the authors incorporate physical information to guide diffusion model. In addition, the authors also conduct detailed analysis on important hyper-parameters and model generalization.

**Weaknesses:**

There are several weaknesses for this current version.
1) The motivation and problem setting are not clear. I would suggest the authors re-organize the "Introduction" to clearly sate the problems as well as the challenges.
2) The overall presentation is a little bit messy.
3) The experiments are only conducted on simulated datasets; how this proposed method would work in a real-world application? For example, what will happen if the low-fidelity CFD data containing nosies such as Gaussian noise/shot noise? Will this proposed method still work?
4) Based on the visual results, the proposed method does not show superior performance compared to other diffusion models.

**Questions:**

I list my questions in "weaknesses" section.

---

> ### Author Response · Authors · 2024-11-20
> **Response to Reviewer Vjib**
>
> We would like to thank the reviewer for the feedback. In what follows, we hope to address any concerns you might have.
>
> ### **[Motivation and Problem]**
> Our work is motivated by the observation that existing works on CFD super resolution typically assume that low-fidelity data is artificially downsampled from high-fidelity sources. However, in real-world scenarios, low-fidelity data is generated by numerical solvers, after which machine learning models upsample it to high-fidelity. This approach aims to produce high-fidelity data more efficiently than direct numerical simulation through numerical solvers.
>
> We address the problem that existing models struggle to recover fine-grained high-fidelity details when evaluated on solver-generated low-fidelity data. To improve reconstruction of detailed and accurate structures, we propose the Importance Weight module to guide the diffusion model during training. We also develop a novel, training-free Residual Correction applied exclusively during inference to ensure physical coherence.
>
> We have revised our manuscript to clearly state the motivation and the problem in the Introduction.
>
> ### **[Real World Applications]**
> Our method aims to accelerate the generation of high-fidelity CFD data by using numerical solves to first generate less expensive low-fidelity data, and then upsample it to high-fidelity using machine learning models. Low-fidelity data generated from less accurate numerical simulation would be the input in real world applications. For fluid dynamic data collected from the real world, there is no dataset with both high-fidelity and low-fidelity pairs. We artificially inject Gaussian noise to our solver generated low fidelity data to mimic real world fluid scenarios. Since the vorticity data are continuous, shot noise is not applicable. We inject N(0,1) noise to 64x64 data and N(0,3) noise to 32x32 data. The experiment results are summarized in the following tables. PG-Diff can still outperform baselines by a margin.
>
> ### Kolmogorov Flow with Gaussian Noise
>
> | Model                 | 4x Upsampling      |                     | 8x Upsampling      |                     |
> |-----------------------|--------------------|---------------------|--------------------|---------------------|
> |                       | **L2**            | **Residual**        | **L2**            | **Residual**        |
> | **Bicubic**           | 3.1657            | 2117.09             | 5.4191            | 8100.92             |
> | **CNN**               | 2.2908            | 1154.44             | 3.1189            | 1416.51             |
> | **GAN**               | 2.9324            | 380.81              | 3.0193            | 1746.95             |
> | **Diffusion**         | 1.7959            | 404.00              | 3.1142            | 281.30              |
> | **Conditional Diffusion** | 1.799         | 229.54              | 3.1380            | 110.21              |
> | **PG-Diff**           | **1.7558**        | **30.2461**         | **2.8986**        | **44.3356**         |
>
> ### McWilliams Flow with Gaussian Noise
>
> | Model                 | 4x Upsampling      |                     | 8x Upsampling      |                     |
> |-----------------------|--------------------|---------------------|--------------------|---------------------|
> |                       | **L2**            | **Residual**        | **L2**            | **Residual**        |
> | **Bicubic**           | 2.2729            | 521.15              | 4.1528            | 4435.18             |
> | **CNN**               | 1.6117            | 187.16              | 2.1800            | 781.93              |
> | **GAN**               | 1.6255            | 439.84              | 2.3673            | 2917.50             |
> | **Diffusion**         | 1.2995            | 51.66               | 2.2624            | 240.42              |
> | **Conditional Diffusion** | 1.3114         | 9.13                | 2.3402            | 53.69               |
> | **PG-Diff**           | **1.2543**        | **6.25**            | **2.0374**        | **5.55**            |
>
> ### **[Visual Results]**
> PG-Diff has demonstrated superior results in reconstructing fine-grained details especially in Kolmogorov Flow and McWilliams Flow. We have updated Figure 3 to highlight the differences between PG-Diff and other diffusion models. Additionally, we also reported LPIPS to measure the perceptual quality of the reconstructed results. PG-Diff outperforms baseline models in this metric.

---

> ### Author Response · Authors · 2024-11-25
> **Follow-up the discussion**
>
> Dear Reviewer Vjib,
>
> We greatly appreciate your time and feedback on our work. We have carefully addressed your comments and clarified potential misunderstandings. Additionally, we also included new experimental results to corroborate our findings.
>
> We kindly invite you to revisit our paper in light of these updates and clarifications. We would greatly appreciate it if you could consider whether our responses warrant a reevaluation of your rating.
>
> Best regards,
>
> The Authors

---

> > ### Author Response · Authors · 2024-11-27
> > **Request for Follow-Up Feedback on Author Rebuttal**
> >
> > Dear Reviewer Vjib,
> >
> > Our detailed rebuttal has been submitted, and we have thoroughly addressed all the points and suggestions you raised. We understand the significant workload involved in reviewing papers, but we kindly request your feedback on our responses to ensure that the discussions are as productive and comprehensive as possible.
> >
> > As the discussion phase is comping to an end, we have so far received feedbacks from many others, and would like to get your insights in refining the final version of our work. We believe there're some misunderstanding and make our claims in the above rebuttal, along with new supporting experiment results. We sincerely hope you can provide us feedbacks.

---

> > > ### Author Response · Authors · 2024-11-30
> > > **Regarding Feedback from Reviewer**
> > >
> > > Dear Reviewer Vjib,
> > >
> > > I hope this message finds you well. We recently submitted our rebuttal and would like to kindly request your feedback on our responses.
> > >
> > > We understand that your schedule is demanding and greatly appreciate the time and effort you dedicate to the review process. Your insights are invaluable to us, and we are eager to address any further questions or concerns you may have.
> > >
> > > Thank you for your attention to this matter. We look forward to your response.
> > >
> > > Best regards,
> > >
> > > Authors

---

> ### Comment · Reviewer_Vjib · 2024-12-02
>
> Thanks so much for your time and effort to address my previous concern. However, the current version still has some limitations in my mind:
>
> 1) The authors mentioned that “Traditional approaches such as Direct Numerical Simulation (DNS) (Orszag, 1970) offer high-resolution solutions. However, they are computationally expensive,...” Based on this statement, it seems that the traditional methods could work well in terms of quality but it may incur additional computational costs. Thus, for experiments, the authors should: (1) include one or two traditional methods; (2) include the computational cost as another performance metric.
>
> 2) I agreed with other reviewers about the statement of “distribution shift”. “Our experiments reveal that state-of-the-art models struggle to recover fine-grained high-fidelity details given solver generated low-fidelity inputs, due to the large distribution shifts.” It is a little bit confusing here, do you mean the large distribution shifts between the “solver generated low-fidelity data” and the “directly downsampled low-fidelity data”? The authors should clarify this and include some evidence. Based on Fig1, it does not seem there are large distribution shifts, instead, it may be only one that includes a little bit more texture information compared to the other?
>
> 3) Thanks the authors for adding the experiments when data contains Gaussian noise. However, adding Gaussian noise may not be sufficient to address my previous concern as the diffusion model may naturally handle Gaussian noise very well due to the “diffusion process” while it may fail if other kinds of noises are injected. It would be better if the authors could simulate a “real-world” scenario with “real-world” noises.

---

### Official Review · Reviewer_EzG5 · 2024-11-10

**Soundness:** 2
**Presentation:** 3
**Contribution:** 2
**Rating:** 3
**Confidence:** 3

**Summary:**

This paper aims to improve on prior super-resolution works by introducing a new importance weighting and training-free correction method to train a diffusion model. The paper presents experiments on 4 benchmarks, and shows improved performance over baselines.

**Strengths:**

The paper has a few key strengths: (1) the model improves over baseline models, (2) the benchmarks are a challenging and convincing set of problems, (3) the DWT-based importance sampling seems to work well, and is well-motivated, (4) the results and ablation studies are well done and provide good insight.

**Weaknesses:**

Unfortunately, this paper also has significant flaws that lead me to doubt its novelty and contributions, and as a result, this paper should be rejected in its current state. The main concern is that many of the claims made in the paper are either not well-supported or are false.

(1) Section 1, Introduction: “We study a novel problem on reconstructing high-fidelity flow fields with solver-generated low-fidelity data, benefiting real-world applications. Our experiments reveal that state-of- the-art reconstruction models fail to generate high-quality outputs due to large distribution shifts between low- and high-fidelity data.”

The first part of this claim, which is central to the paper, is not true. Upsampling solver-generated low-fidelity data has been both posed as well as solved before [1, 2]. While the provided references are not exhaustive, prior work has considered the limitation of super-resolution works downsampling high-resolution data and have worked directly with solver-generated low-fidelity data.

The second part of this claim does not seem to be well-supported. While it is true that the proposed model outperforms current diffusion models, there is not much evidence for the provided reason being due to a distribution shift. In fact, one of the main insights presented in the cited prior work from Shu et al. [3] is that by noising data samples, they all approach the same distribution regardless of the original distribution being from high or low fidelity data. By that argument, diffusion models should be agnostic to whether the guide data comes from downsampled high-resolution simulations or low-resolution simulations, since when fully noised these samples are drawn from nearly identical distributions.

(2) Section 4.6, Model Generalization: “We observe that PG-Diff generalizes well even beyond its training distribution”

The spatial domain size variations don’t seem to generate samples outside of the training distribution, since the smaller domains are a subset of the larger training domain. The other test cases could be out of the training distribution, but even then, using a smaller dt shouldn’t significantly alter the data distribution.

(3) Section 4.3, Runtime Comparison:“PG-Diff is considerably faster than the time required to produce high- fidelity data directly through numerical solver”

Following one of the cited papers from McGreivy & Hakim [4], the runtime comparison with a numerical solver is not faithfully done. In particular, the numerical solver should be coarsened until it achieves a similar L2 error as the proposed method. The runtime of the coarsened solver should be compared with the proposed method instead of the zero-error, ground truth numerical runtime.

There are a few more minor concerns about the work.

(1) Section E.4: The argument about LPIPS being a suitable metric is not empirically or theoretically sound. The given reasoning about ImageNet containing multiscale features and textures leading to it generalizing to CFD applications is somewhat hand-wavy and without providing evidence, it is challenging to claim that a model trained on ImageNet would be “a robust metric” for physics simulations.

(2) Multiscale Evaluation: The authors claim to achieve best or near-best results on all subdomains but there are results where the proposed method is not the best, especially in the Appendix Figures 11, 12, 13.

(3) Residual Correction During Inference: While the technique improves performance, it seems highly dependent on hyperparameters. In particular, there is reduced performance when using more correction steps, likely due to the data being corrected to samples outside of the model’s training distribution. It seems odd that a core method of the paper asymptotically reduces performance (i.e., as more compute/refinement is done, the worse the model gets).

**Questions:**

Do you have any intuition as to why the model trained on Re=2000 performs worse than the model trained on the original data when evaluated on a Re=2000 test case? (Table 3)
Is there a reason why the bicubic upsampling is quite slow? (Table 8)

As a whole, the main contributions of the paper seem to be an importance weighting and correction step during inference, which seems to be an incremental improvement on a prior work from Shu et al. [3].

Rajat Kumar Sarkar, Ritam Majumdar, Vishal Jadhav, Sagar Srinivas Sakhinana, Venkataramana Runkana. Redefining Super-Resolution: Fine-mesh PDE predictions without classical simulations. https://arxiv.org/abs/2311.09740
Francis Ogoke, Quanliang Liu, Olabode Ajenifujah, Alexander Myers, Guadalupe Quirarte, Jonathan Malen, Jack Beuth, Amir Barati Farimani. Inexpensive high fidelity melt pool models in additive manufacturing using generative deep diffusion. https://www.sciencedirect.com/science/article/pii/S0264127524005562
Dule Shu, Zijie Li, Amir Barati Farimani. A physics-informed diffusion model for high-fidelity flow field reconstruction. https://www.sciencedirect.com/science/article/pii/S0021999123000670
Nick McGreivy, Ammar Hakim. Weak baselines and reporting biases lead to overoptimism in machine learning for fluid-related partial differential equations. https://www.nature.com/articles/s42256-024-00897-5

---

> ### Author Response · Authors · 2024-11-20
> **Response to Reviewer EzG5**
>
> We would like to thank the reviewer for the feedback. In what follows, we hope to address any concerns you might have.
>
> ### **[Introduction Statement]**
> Thank you for pointing out these related works. We have revised our manuscript to correctly cite and discuss these works. However, we would like to point out that even though [1, 2] benchmarks their super resolution models on solver generated low-fidelity data, they did not explicitly address the loss of fine-grained details in reconstructed samples under such a setting. [1] combines a Unet with physics-informed loss function, and [2] uses a CNN and standard diffusion models. Our paper, on the other hand, proposes novel Importance Weight and Residual Correction to enhance the predictive accuracy and physical consistency of the reconstructed samples.
>
> The state-of-the-art diffusion model is unable to reconstruct fine-grained details because solver generated low-fidelity data contains less information on these fine-grained details compared to downsampled low-fidelity data. Thus, we propose Importance Weight to force the diffusion model to reconstruct fine-grained details at different noise levels during training.
>
> [3] only implies that adding noise would make downsampled low-fidelity data and noisy high fidelity data to approach the same distribution. However, with significant distribution shifts, for example, Kolmogorov Flow data with McWilliams Flow data, we would need to add a significant amount of noise to make these two distributions similar. At that point, both noised distributions would be completely random, and the diffusion model generates data unconditionally.
>
> We have updated our manuscript to clearly express these thoughts.
>
> ### **[Section 4.6 Model Generalization Experiment]**
> Our super resolution model is aimed to combine with traditional numerical solvers to accelerate high-fidelity simulation. Thus, we aim to test whether PG-Diff, trained with data from one solver configuration, can be extended to data from different solver configurations. We do not claim that PG-Diff can generalize to distributions significantly different from training.
>
> ### **[Runtime Comparison]**
> Our runtime comparison is revised under the new setting. Results show that PG-Diff can still accelerate the generation of high-fidelity data.
>
> ### **[LPIPS Score]**
> We believe that the perceptual aspects of LPIPS are not inherently tied to ImageNet’s specific domain (natural images) but arise from the hierarchical feature extraction of convolutional architectures. Low-level features such as edges and textures, and mid-level features such as structures and patterns, are domain-agnostic. In the case of CFD vorticity data, we observe analogous structures and patterns, such as coherent vortical regions and flow features.
>
> We also add PSNR and SSIM results to demonstrate the superior performance of PG-Diff in the Appendix.
>
> ### **[Multiscale Evaluation]**
> In Line 375, we claimed that “PG-Diff demonstrates superior performance in the LL, LH, and HL subdomains, achieving the best or near-best results among all methods”. We acknowledge that PG-Diff does not have best performance in HH subdomain, and we explicitly stated that. For LL, LH, and HL subdomains, our claim that PG-Diff has the best or near-best results among all methods does hold true.
>
> ### **[Residual Correction]**
> The hyperparameter tuning of our residual correction is conducted on Kolmogorov Flow only, and the same hyperparameter is used for all other datasets. Our Residual Correction involves multiple steps of Adam gradient descent applied at selected backward diffusion steps. The gradient descent projects the reconstructed samples onto the solution subspace of the PDE by minimizing the residuals. However, in super resolution tasks, we not only want the reconstructed sample to lie on the solution subspace of the PDE to ensure physical consistency, we also want the difference between reconstructed sample and ground truth high-fidelity data to be minimized to ensure predictive accuracy. Thus, we conduct experiments in Section 4.5 PHYSICAL GUIDANCE to find the optimal scheduler and number of correction steps to ensure optimal balance between physical consistency(Residual) and predictive accuracy(L2 Loss).
>
> ### **[Generalization Experiment Re=2000]**
> Due to computational limitations, the models in generalization experiments are only trained for one time. The improvements likely come
> from model trained on Re=2000 data being an outlier.

---

> > ### Author Response · Authors · 2024-11-20
> > **Response to Reviewer EzG5 Continued**
> >
> > ### **[Bicubic Interpolation Runtime]**
> > The original bicubic interpolation was tested on cpus. We have revised Table 8 to report the runtime of pytorch implemented bicubic interpolation to run on gpus. We would like to comment that since the predictive accuracy, physical consistency, and perceptual quality of bicubic interpolation reconstructed samples are significantly worse than PG-Diff and other baselines, our results that PG-Diff is better than the baselines still hold true.
> >
> > [1] Redefining Super-Resolution: Fine-mesh PDE predictions without classical simulations.\
> > [2] Inexpensive high fidelity melt pool models in additive manufacturing using generative deep diffusion.\
> > [3] A physics-informed diffusion model for high-fidelity flow field reconstruction.

---

> > > ### Author Response · Authors · 2024-11-25
> > > **Follow-up the discussion**
> > >
> > > Dear Reviewer EzG5,
> > >
> > > We greatly appreciate your time and feedback on our work. We have carefully addressed your comments and clarified potential misunderstandings. Additionally, we also included new experimental results to corroborate our findings.
> > >
> > > We kindly invite you to revisit our paper in light of these updates and clarifications. We would greatly appreciate it if you could consider whether our responses warrant a reevaluation of your rating.
> > >
> > > Best regards,
> > >
> > > The Authors

---

> > > > ### Author Response · Authors · 2024-11-27
> > > > **Request for Follow-Up Feedback on Author Rebuttal**
> > > >
> > > > Dear Reviewer EzG5,
> > > >
> > > > Our detailed rebuttal has been submitted, and we have thoroughly addressed all the points and suggestions you raised. We understand the significant workload involved in reviewing papers, but we kindly request your feedback on our responses to ensure that the discussions are as productive and comprehensive as possible.
> > > >
> > > >
> > > > As the discussion phase is comping to an end, we have so far received feedbacks from many others, and would like to get your insights in refining the final version of our work. We believe there're some misunderstanding and make our claims in the above rebuttal. We sincerely hope you can provide us feedbacks.

---

> > > > > ### Author Response · Authors · 2024-11-30
> > > > > **Regarding Feedback from Reviewer**
> > > > >
> > > > > Dear Reviewer EzG5,
> > > > >
> > > > > I hope this message finds you well. We recently submitted our rebuttal and would like to kindly request your feedback on our responses.
> > > > >
> > > > > We understand that your schedule is demanding and greatly appreciate the time and effort you dedicate to the review process. Your insights are invaluable to us, and we are eager to address any further questions or concerns you may have.
> > > > >
> > > > > Thank you for your attention to this matter. We look forward to your response.
> > > > >
> > > > > Best regards,
> > > > >
> > > > > Authors

---

### Comment · Area_Chair_8FEW · 2024-11-25
**Reviewers' Response**

Dear Reviewers,

As the author-reviewer discussion period is approaching its end, I would strongly encourage you to read the authors' responses and acknowledge them, while also checking if your questions/concerns have been appropriately addressed.

This is a crucial step, as it ensures that both reviewers and authors are on the same page, and it also helps us to put your recommendation in perspective.

Thank you again for your time and expertise.

Best,

AC

---

### Meta-Review · Area_Chair_8FEW · 2024-12-18

**Metareview:**

This paper proposes PG-Diff, a diffusion model for super-resolution in CFD, that reconstructs high-fidelity flow fields from low-fidelity inputs. The key innovations include an "Importance Weight" strategy during training, prioritizing high-frequency components via wavelet transforms, and a training-free "Residual Correction" method during inference, incorporating physical constraints from governing equations like the Navier-Stokes equations. The authors evaluated PG-Diff on four CFD benchmarks, where it demonstrated improved accuracy and reduced residuals compared to existing super-resolution methods.

The reviewers generally agree that the paper is well-written and easy to follow, highlighting the use of wavelet transforms and physics-informed (residual) corrections to enhance the performance of the diffusion model in CFD benchmarks with respect to the baselines.

However, many reviewers raised questions about the validity of the claims, particularly with respect to the issue of distribution shifts and the claims on speed-up with respect to numerical solvers. The reviewers also raised concerns about the positioning of the paper in the literature regarding the super-resolution of divergent solutions (also called downscaling), noting that the problem has already been studied and tackled before; particularly using diffusion models in [1,2,3]. Thus, greatly reducing the novelty of the approach, as many of those baselines are missing.

Given the weaknesses of the paper, I recommend rejection.

References:

[1] Mardani, Morteza, et al. "Residual corrective diffusion modeling for km-scale atmospheric downscaling, 2024." URL https://arxiv. org/abs/2309.15214.

[2] Bischoff, Tobias, and Katherine Deck. "Unpaired downscaling of fluid flows with diffusion bridges." Artificial Intelligence for the Earth Systems 3.2 (2024): e230039.

[3 ]Wan, Zhong Yi, et al. "Debias coarsely, sample conditionally: Statistical downscaling through optimal transport and probabilistic diffusion models." Advances in Neural Information Processing Systems 36 (2023): 47749-47763.

**Additional Comments On Reviewer Discussion:**

The main issue raised by the reviewers was that the claims were not properly backed up by numerical experiments, and the authors were not familiar with an extensive literature solving the same problem (including methods leveraging diffusion models). The authors did not address these issues satisfactorily.

---

### Decision · Program_Chairs · 2025-01-22

Reject